# Histone variant H3.3 residue S31 is essential for *Xenopus* gastrulation regardless of the deposition pathway

David Sitbon [1,3], Ekaterina Boyarchuk [1,3], Florent Dingli [2], Damarys Loew [2] & Geneviève Almouzni [1✉]

Vertebrates exhibit specific requirements for replicative H3 and non-replicative H3.3 variants during development. To disentangle whether this involves distinct modes of deposition or unique functions once incorporated into chromatin, we combined studies in *Xenopus* early development with chromatin assays. Here we investigate the extent to which H3.3 mutated at residues that differ from H3.2 rescue developmental defects caused by H3.3 depletion. Regardless of the deposition pathway, only variants at residue 31—a serine that can become phosphorylated—failed to rescue endogenous H3.3 depletion. Although an alanine substitution fails to rescue H3.3 depletion, a phospho-mimic aspartate residue at position 31 rescues H3.3 function. To explore mechanisms involving H3.3 S31 phosphorylation, we identified factors attracted or repulsed by the presence of aspartate at position 31, along with modifications on neighboring residues. We propose that serine 31-phosphorylated H3.3 acts as a signaling module that stimulates the acetylation of K27, providing a chromatin state permissive to the embryonic development program.

[1] Institut Curie, PSL Research University, CNRS, Sorbonne Université, Nuclear Dynamics Unit, Equipe Labellisée Ligue contre le Cancer, 26 rue d'Ulm, 75005 Paris, France. [2] Institut Curie, PSL Research University, Centre de Recherche, Mass Spectrometry and Proteomics Facility (LSMP), 26 rue d'Ulm, Paris 75248 Cedex 05, France. [3] These authors contributed equally: David Sitbon, Ekaterina Boyarchuk. ✉email: genevieve.almouzni@curie.fr

The organization of DNA into chromatin provides not only a means for compaction but also a versatile landscape contributing to cell fate and plasticity[1,2]. The basic unit of chromatin, the nucleosome core particle, is composed of a histone tetramer $(H3-H4)_2$ flanked by two histone dimers H2A–H2B, around which 147 bp of DNA is wrapped[3]. Functional modulation of the nucleosome occurs though the choice of histone variants and reversible posttranslational modifications (PTMs) such as acetylation, methylation, and phosphorylation[4]. Three out of the four core histone families possess histone variants[5–7]. Within the H3 family of histones, the centromere-specific CENP-A is distinct, whereas the other well-characterized H3 variants are closely related and are thought to function similarly[8]. Those H3 variants exhibit similar structural features at the core particle level[9]; however, they display important differences in their cell cycle regulation and modes of incorporation into chromatin[10].

To date, two distinct modes of incorporation have been described: DNA-synthesis independent (DSI) and DNA-synthesis coupled (DSC); the latter being used during both DNA replication and repair[11]. In humans, the two replicative variants H3.1 and H3.2 are incorporated into chromatin in a DSC manner via the histone chaperone complex CAF-1[12–18]. The non-replicative form H3.3, which differs from H3.1 and H3.2 by five and four residues, respectively, is incorporated in a DSI manner[17,19–21]. DSI incorporation depends on the HIRA histone chaperone complex in euchromatin regions[17,19,21,22], whereas the presence of H3.3 at telomeres and pericentric heterochromatin relies on the DAXX/ATRX histone chaperone complex[22–24]. Thus, the dynamics of the different histone variants in regards to their deposition is linked to dedicated histone chaperones[25,26].

Inspired by the quote "nothing in biology makes sense except in the light of evolution[27]," we considered H3 variants in light of their conservation in different organisms. In *Saccharomyces cerevisiae*, there is only one non-centromeric histone H3 (which is most related to human H3.3) that provides both essential replicative and non-replicative variant functions[28–30]. Paradoxically, however, in humanized *S. cerevisiae* strains where all histones are exchanged for human orthologs, replacement with hH3.1 more readily produced colonies than with the hH3.3 variant[31]. In metazoans such as *Drosophila melanogaster*, the replicative variant can compensate for the loss of H3.3 during development in somatic tissues, although the adults are sterile[32–36]. As sterility could simply reflect a shortage of maternal H3.3 to replace protamine from sperm chromatin after fertilization, the most parsimonious hypothesis suggests that the nature of the variant itself might not be critical. Similarly, H3.3 is not essential in *Caenorhabditis elegans*, where its removal is not lethal but reduces fertility and viability in response to stress[37]. However, in *Arabidopsis thaliana*, replicative and non-replicative H3 variants are essential. The absence of H3.3 leads to embryonic lethality and also partial sterility due to defective male gametogenesis[38].

In mouse, the deletion of one of the two copies of the H3.3 gene results in developmental defects at E12.5 and sterility[39–41]. In human, dominant effects of substitutions in H3.3, such as H3.3 K27M and H3.3 G34R/V, along with mutations affecting their chaperones such as DAXX/ATRX, have been implicated in different types of cancer[42–49]. Thus, the developmental defects observed mouse models and mutations associated with particular cancers underline the importance of individual histone H3 variant and their chaperones in vertebrates.

Given the high degree of sequence identity between H3 variants, whether the need for a particular histone variant could reflect either (i) a unique mode of incorporation and provision or (ii) a distinct identity once incorporated into chromatin to drive their functions remains puzzling. Although the first hypothesis has largely been favored based on previous work, including our own, the issue has not been formally addressed. To disentangle these two possibilities, we decided to use the *Xenopus laevis* model, as it represents an ideal system to tackle such issue. Indeed, extensively characterized both in developmental biology and chromatin studies[50–52], its external development permits direct access to embryos for observation and manipulation[53]. With retention of H3 variants in sperm[54] and only one replicative histone (H3.2), it provides an ideal situation while retaining amino acid sequence conservation with human variants for both H3.2 and H3.3. Following fertilization, *X. laevis* development starts with 12 rapid embryonic cell divisions, which include only S and M phases[55–57]. At the midblastula transition (MBT), zygotic activation occurs concomitantly with a progressive lengthening of the cell cycle, to reach a typical cell cycle with two gap phases at gastrulation. In addition, cells begin to differentiate with the acquisition of migration properties. Importantly, previous work in our laboratory revealed a specific requirement for H3.3 during *X. laevis* early development at the time of gastrulation[58]. That work demonstrated that depletion of endogenous H3.3 leads to severe gastrulation defects that cannot be rescued by providing the replicative counterpart H3.2. Interestingly, there are only four residues that differ between H3.2 and H3.3. A first region of divergence within the AIG motif—in the globular domain of H3.3 —is involved in histone variant recognition by dedicated histone chaperones[59–62]. The other distinct residue, a serine only present in H3.3, is located on the histone N-terminal tail at position 31 and can be phosphorylated[63–65].

Here we systematically mutate the H3.3 histone variant at each of its distinct residues, to assess their ability to rescue the gastrulation defects and examine their mode of chromatin incorporation. We find that mutations affecting the incorporation pathway are neutral in setting specific H3.3 functions at the time of gastrulation. In contrast, serine 31 is critical to rescue defects following endogenous H3.3 depletion. In *Xenopus*, Ser31 —which is conserved in multicellular organisms including humans—is phosphorylated with a peak in mitosis by a network of mitotic kinases, including CHK1 and Aurora B. Remarkably, a phospho-mimic form of H3.3, S31D (which cannot be dynamically modified) still fully rescues gastrulation. Interestingly, analysis of protein interactions and repulsions on the phospho-mimic peptide, as shown by mass spectrometry, reveals attraction of transcription cofactors as those involved in the β-catenin pathway in interphase, but repulsion of factors involved in chromosome condensation and splicing in mitosis. We also find that H3.3 S31D exhibits an increase in H3.3K27ac and a loss of H3.3K27me3 in-*cis*. We discuss how this evolutionarily conserved residue conveys, in both interphase and mitosis, unique properties for the H3.3 variant in vertebrates during cell cycle and cell-fate commitment.

## Results

**H3.3 dosage is critical for *X. laevis* gastrulation.** Although *Homo sapiens* have two replicative H3 variants H3.1 and H3.2, *X. laevis* only possesses one replicative variant H3.2. Both H3.2 and H3.3 are conserved with their human orthologs (Fig. 1a). Interestingly, the two H3 variants are almost identical and conserved through evolution[66]. Two regions show differences in H3.2 and H3.3. The first one encompasses positions 87, 89, and 90 with a serine, a valine, and a methionine, known as the SVM motif in H3.2. Instead, these positions correspond to an alanine, an isoleucine, and a glycine, known as the AIG motif in H3.3. The second difference lies at position 31 where H3.2 shows an alanine and H3.3 a serine. Considering the sequences for H3.2 and H3.3 histone variants from five different model organisms, in which the functions along with deposition pathways of H3 variants have

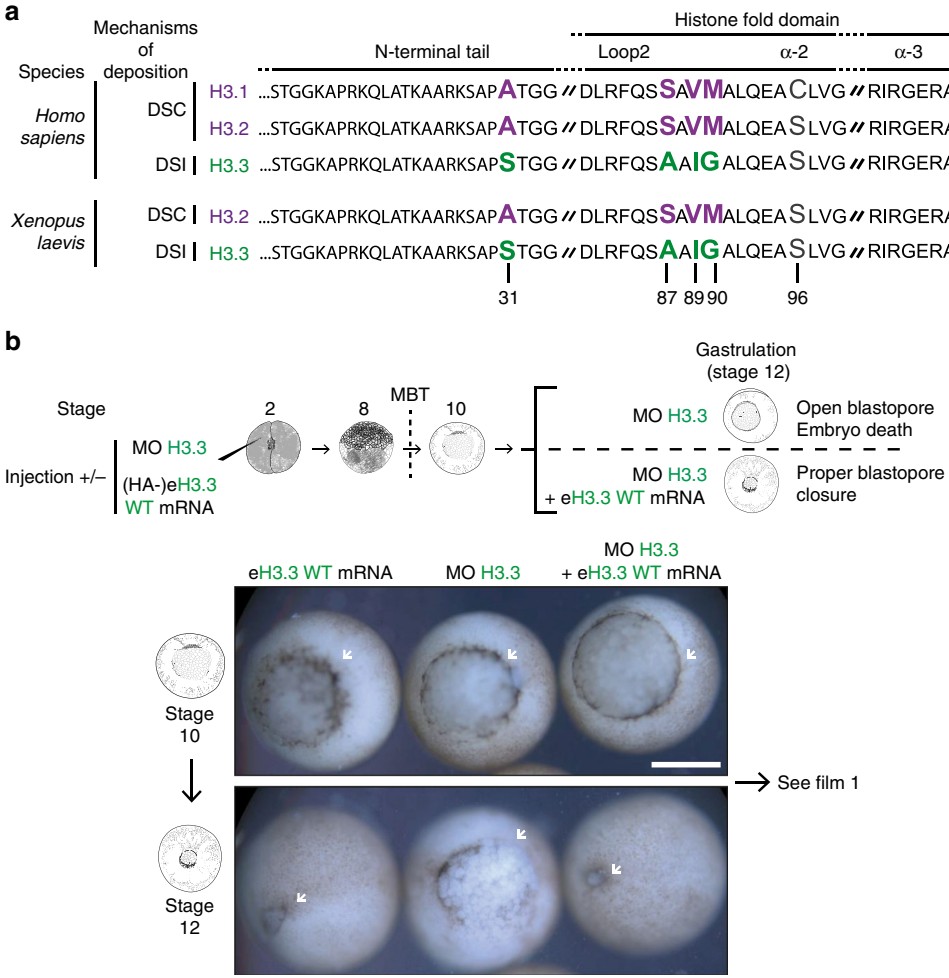

**Fig. 1 H3.3 is essential for gastrulation of *X. laevis*. a** Best-studied non-centromeric H3 histone variants in *H. sapiens* and *X. laevis*. The two well-characterized forms of non-centromeric H3 variants correspond to the replicative histones H3.1 and H3.2, and the non-replicative histone H3.3, depicted in purple and green, respectively. In humans, the replicative H3 variants differ by only one residue at position 96, a cysteine and a serine, respectively. The non-replicative form H3.3 shares more than 96% identity with the replicative forms, with five and four residue differences with H3.1 and H3.2, respectively. In addition, *X. laevis* embryos possess only one replicative histone variant, H3.2. Finally, histone sequences are conserved between *H. sapiens* and *X. laevis*. DSC: DNA-synthesis coupled, DSI: DNA-synthesis independent. **b** H3.3 depletion-associated defects and rescue. Morpholino and/or eH3.3 mRNA are injected at the two-cell stage and associated defects can be observed at the gastrulation stage if not rescued with eH3.3 WT. White arrowheads indicate the blastopore closure. Scale bar corresponds to 500 μm. See Supplementary Movie 1.

been studied, the replicative variant H3.2 exhibit ~3% dissimilarity (four variable residues out of 136; Supplementary Fig. 1a). In the case of the non-replicative variant H3.3, it varies by 4%, with six variable residues. Remarkably, the region responsible for histone chaperone recognition shows the highest variation, in line with possible coevolution with their respective histone chaperones that are not as closely conserved (Supplementary Fig. 1b). To examine how deposition pathways and the role for histone variants are related, targeting these regions could thus be considered. Using *X. laevis* embryos, we had previously used a morpholino specifically designed to target endogenous H3.3[58] (see Methods). We found that this morpholino against H3.3 leads to defects during late gastrulation (Fig. 1b and Supplementary Movie 1). Although the blastopore forms and invaginates during gastrulation, depletion of endogenous H3.3 leads to an arrest of the blastopore closure. When co-injected with the morpholino, exogenous hemagglutinin (HA)-tagged H3.3 mRNAs (hereafter referred to as eH3s), but not HA-tagged H3.2, can rescue the phenotype[58]. To better define the relationship between endogenous H3.3 and its functional importance during development, we

decided to titrate the concentration of morpholino (Supplementary Fig. 1c). The lower concentration enabled the blastopore to start to invaginate without complete closure and led to late gastrulation defects, consistent with our previous findings[58]. By increasing by two- to fourfold morpholino concentration, gastrulation defects coincided with an even earlier phenotype, where the blastopore closure did not occur at all. These data are in line with a titration effect, whereby gastrulation time allows to readily reveal requirements for H3.3. We could thus use it as a readout to assess the need for a distinct mode of incorporation for H3 variants or for the histone variant itself once incorporated.

**Swapping deposition mode retains H3.3 roles at gastrulation.** To investigate the importance of the deposition pathway, we first considered the H3.3 histone chaperone recognition motif (Fig. 2a). Incorporation into chromatin of the non-replicative variant H3.3 occurs throughout cell cycle (DSI) and involves the HIRA complex in gene-rich regions[17,19–21] and the DAXX/ATRX complex in heterochromatin regions[22–24,67,68]. In contrast, incorporation of the replicative variants is coupled to DNA

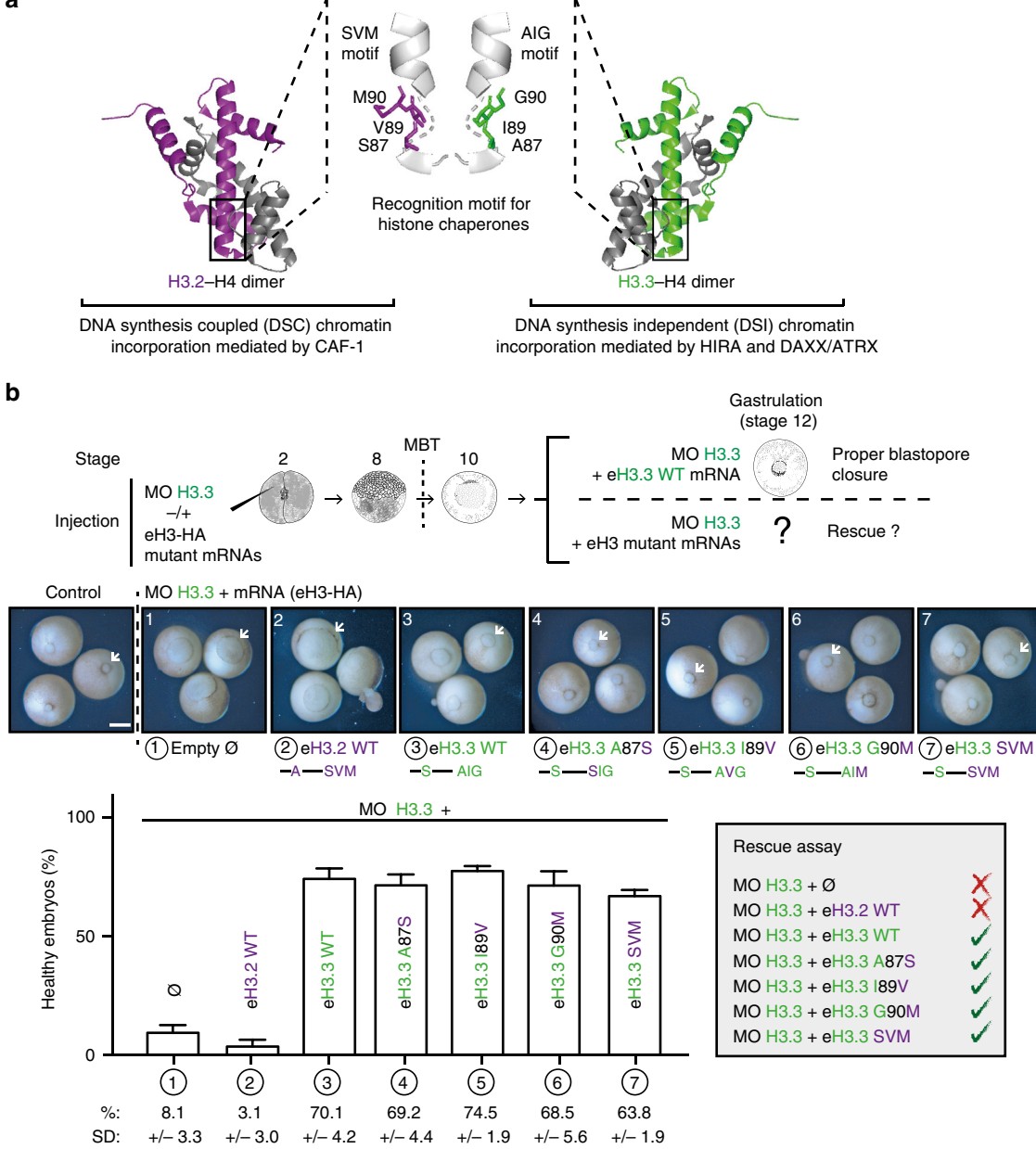

**Fig. 2 eH3.3 AIG mutants rescue depletion of H3.3 during *X. laevis* early development. a** Highlights of the histone chaperone recognition motif residues of H3 variants. Dedicated histone chaperones recognize histone variants by the H3.2 SVM and H3.3 AIG motifs. Although both motifs are structurally similar, the main difference appears for the residue 90. Crystal structures adapted from PDB ID codes 5B0Z[105] and 3 AV2[9]. **b** Rescue assays with H3.3 AIG mutants. Injections are performed at two-cell stage. Effect on gastrulation is analyzed at stage 12. White arrowheads indicate the blastopore closure. Scale bar corresponds to 500 μm. Bar plot shows mean ± SD for three independent experiments with more than 30 embryos. Source data are provided as a Source Data file.

synthesis (DSC) and is mediated by the CAF-1 complex[12–18]. Importantly, structural studies enabled the identification of how histone chaperones discriminate the distinct histone variants through a motif located in the globular domain of histones[59–61]. Both DAXX and HIRA complexes bind to the AIG motif of H3.3, with a particular affinity for the glycine at position 90. Therefore, we tested the ability of mutated H3.3 in the histone chaperone recognition motif to rescue the loss of H3.3 (Fig. 2b). Single mutants for each residue of the AIG motif, i.e., eH3.3 A87S, eH3.3 I89V, and eH3.3 G90M, could rescue loss of endogenous H3.3 and embryo development occurred with the same efficiency as eH3.3 WT (wild type), whereas eH3.2 WT could not. As the individual substitution in the H3.3 motif did not affect the

developmental rescue in vivo, we assessed whether mutation of all three residues of the motif would then affect H3.3 functions. To our surprise, in this context, eH3.3 SVM hybrid form proved still able to rescue the loss of endogenous H3.3. In addition, such mutant forms were expressed and incorporated into chromatin at similar levels in the embryo (Supplementary Fig. 2a, b). Thus, by substituting the H3.3 recognition motif for its histone chaperone with the one from H3.2, we could ensure the rescue in vivo. As early development may allow looser interactions with the dedicated chaperones compared with a somatic context, we decided to assess both interaction and incorporation means with mutated H3.3 for histone chaperone recognition motif. eH3.3 carrying the H3.2 recognition motif is able to rescue the depletion of H3.3,

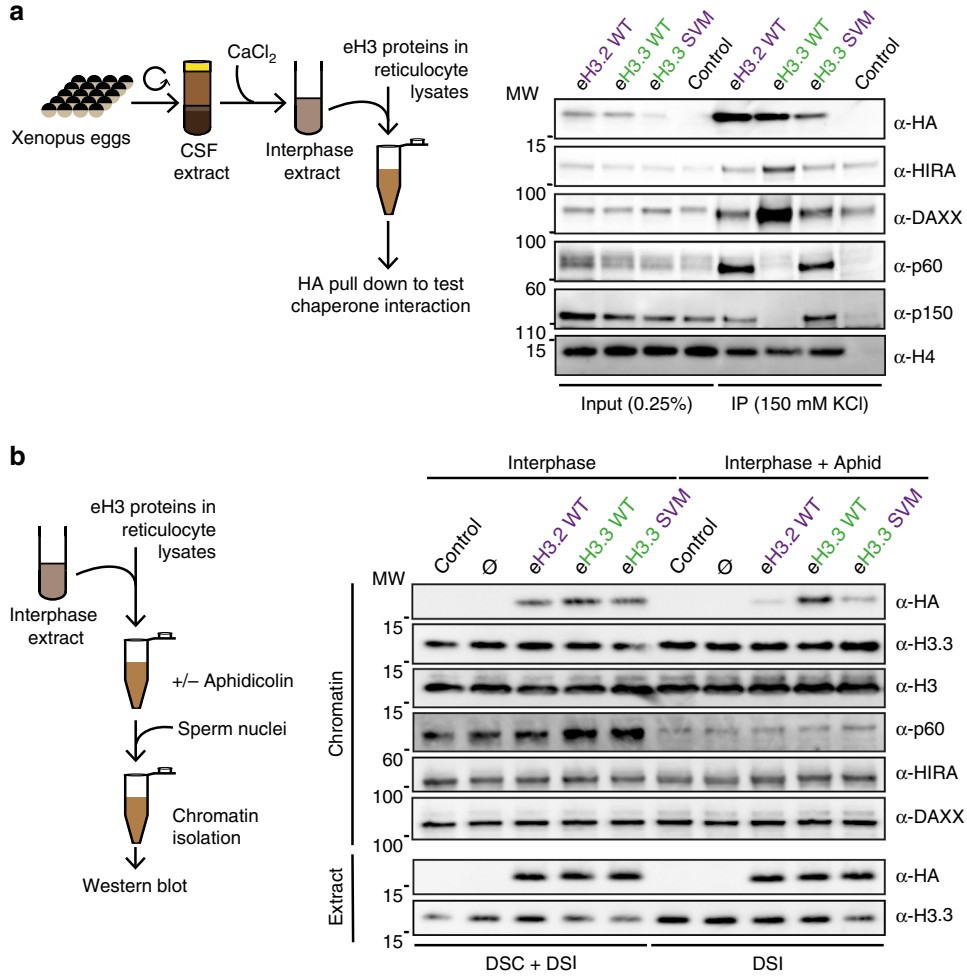

**Fig. 3 H3.3 AIG motif permutation changes histone chaperone interactions and histone modes of incorporation into chromatin in *X. laevis*.**
**a** Immunoprecipitation of eH3 AIG triple mutant and controls in interphase extracts. Recombinant proteins are produced in rabbit reticulocyte lysates and pulled down by their HA-tag after incubation. The binding of HIRA, DAXX, p60, and p150 is analyzed by western blotting with specific antibodies. α-HA antibody is used to detect eH3.3. H4 is used as an internal control. Primary antibodies indicated on the right. Molecular weight (MW) markers in kDa indicated on the left. **b** Incorporation of eH3 AIG triple mutant and controls into sperm chromatin in interphase extracts. Purified nuclei remodeled in the interphase extracts supplemented with indicated eH3.3 in the presence or absence of aphidicolin are analyzed by western blotting with indicated antibodies. α-HA antibody is used to detect eH3.3. Primary antibodies indicated on the right. Molecular weight (MW) markers in kDa indicated on the left. DSC: DNA-synthesis coupled, DSI: DNA-synthesis independent mode of incorporation. Source data are provided as a Source Data file.

suggesting that this hybrid form can still ensure H3.3 functions during early development. In this context, whether the histone chaperone recognition motif actually determined the respective deposition pathways for each histone variant remained unknown. First, we analyzed the chaperone interactions with the eH3.3 SVM hybrid form by immunoprecipitations (Fig. 3a). Both H3.3 dedicated chaperone complexes HIRA and DAXX recognized best eH3.3 WT, carrying the H3.3 AIG motif. However, p60 and p150, two subunits of the CAF-1 complex dedicated to H3.2, could recognize eH3.2 WT and eH3.3 SVM equally, but not eH3.3 WT. This shows that eH3.3 SVM can be recognized by CAF-1, arguing for a possible swap in the means for incorporation. We confirmed this finding in vivo by immunoprecipitating the various mutants of eH3 directly from embryos at the gastrulation stage (Supplementary Fig. 3a, b). Interestingly, H3.3 dedicated chaperones did recognize H3.3 variants with a single mutation of the AIG motif in vivo. Therefore, the whole recognition motif is key to alter chaperone interactions. We then explored the potential impact of eH3 mutants on the mode of histone variant incorporation. To test this, we performed chromatin assembly assays using extracts derived from *X. laevis*

eggs[52,69]. We supplemented *Xenopus* egg extracts with eH3.2 WT, eH3.3 WT, or eH3.3 SVM, and monitored their incorporation into chromatin using sperm nuclei under conditions allowing or preventing DNA synthesis (Fig. 3b). In interphase extracts, sperm DNA forms nuclei and can replicate and reassemble chromatin, whereas mitotic extracts lack DNA replication capacity. We isolated and analyzed sperm chromatin nuclei from interphase extracts in the presence or absence of the DNA synthesis inhibitor aphidicolin. Incorporation of eH3.3 WT using sperm nuclei occurred with a similar efficiency in the presence or absence of DNA synthesis. By contrast, eH3.2 WT incorporation was severely diminished by the presence of aphidicolin. Importantly, eH3.3 SVM incorporation showed the same dependency on DNA replication, arguing that its incorporation mode switched toward a DSC mechanism. Consistently, p60 recruitment to chromatin is, as expected, highly reduced when DNA synthesis is inhibited. We further confirmed in mitotic extracts that only the variant with the AIG motif could get incorporated independently of DNA synthesis (Supplementary Fig. 3c). Based on these data, we conclude that the H3.3 histone variant was efficiently provided regardless of the incorporation pathway. Thus, defects at the time

of gastrulation do not arise from a need for a distinct incorporation pathway outside S phase independently of DNA synthesis. It rather reflects an inherent feature of the variant when incorporated into chromatin. We thus examined more closely if the only the remaining specific residue of H3.3, the amino acid 31, could account for this unique feature.

**H3.3 S31 is critical and phosphorylated by CHK1 and Aurora B**. To address the role of the H3.3 residue at position 31, we first constructed a new H3.3 mutant—H3.3 S31A—containing an alanine instead of a serine at position 31, while maintaining its

original AIG motif (Fig. 4a and Supplementary Fig. 4a). eH3.3 S31A did not rescue endogenous H3.3 depletion during *Xenopus* early development. Therefore, H3.3 serine at position 31 cannot be substituted by an alanine (which is the corresponding residue in H3.2) to fulfill H3.3 dedicated functions in this time window during development. Interestingly, this particular residue has been found phosphorylated in human cells during mitosis[63]. Furthermore, a threonine substitutes for this serine in *Arabidopsis*, a residue that might possibly also undergo phosphorylation, something that has however not yet been documented. A key question is thus whether the actual need for H3.3 is linked to the capacity of H3.3 S31 to become phosphorylated. To this end,

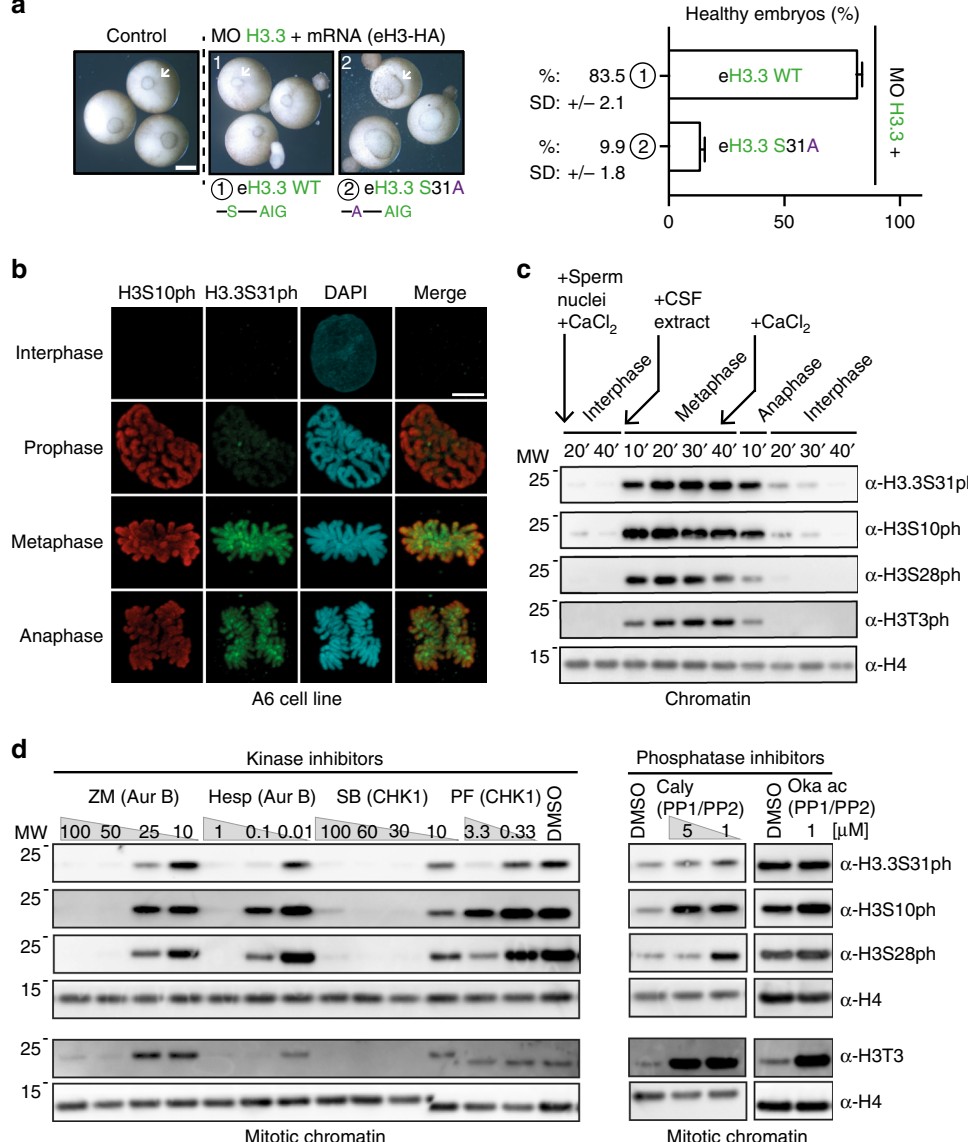

**Fig. 4 In *X. laevis*, H3.3 S31 is critical for early development and is phosphorylated. a** Rescue assays with eH3.3 S31A mutant. Injections are performed at two-cell stage. Effect on gastrulation is analyzed at stage 12. White arrowheads indicate the blastopore closure. Scale bar corresponds to 500 μm. Bar plot shows mean ± SD for three independent experiments with more than 30 embryos. **b** Representative immunofluorescence images showing 3D-distribution and timing of H3.3S31 and H3S10 phosphorylation in the *Xenopus* A6 cell line. Scale bar represents 10 μm. **c** H3.3 S31ph dynamics in *Xenopus* egg extracts. Nuclei remodeled in extracts cycled to interphase, mitosis, and then to the second interphase are purified at the indicated time. Kinetics (appearance and removal) of the H3 phosphorylation is analyzed by western blotting with indicated antibodies. H4 is used as a loading control. Primary antibodies indicated on the right. Molecular weight (MW) markers in kDa indicated on the left. **d** Effects of kinase and phosphatase inhibitors on H3 phosphorylation in mitotic egg extracts. Purified nuclei remodeled in the CSF-arrested extracts supplemented with indicated inhibitors are analyzed by western blotting with indicated antibodies. H4 is used as a loading control. Primary antibodies indicated on the right. Molecular weight (MW) markers in kDa indicated on the left. Source data are provided as a Source Data file.

we first examined H3.3 S31 phosphorylation in the *Xenopus* system using a *Xenopus* A6 cell line derived from the kidney (Fig. 4b). By immunofluorescence, we detected a strong enrichment of H3.3 S31ph during mitosis. Interestingly, when compared with another mitotic modification common to both H3 variant forms, H3 S10ph, its pattern was different, indicating possibly a distinct function. Although H3 S10ph covers the edges of mitotic chromosomes, H3.3 S31ph is enriched at a centric and pericentric heterochromatin. This pattern, similar to previous observations in HeLa B cells (Supplementary Fig. 4b) as in ref. [63], is consistent with observations in mouse and monkey cell lines[64,65]. We further characterized the acquisition of this mark in interphase extract or in extract pushed into mitosis (Supplementary Fig. 4c). We did not detect any significant signal for H3.3 S31ph in the soluble pool of H3.3 in either mitotic or interphase extracts, indicating that the modification is likely acquired once the variant is incorporated into chromatin. In addition, we could not detect a signal for this mark on non-remodeled sperm chromatin or chromatin assembled in interphase extracts. This mark was mostly enriched in the fraction corresponding to isolated mitotic chromatin, in a profile resembling the H3 S10ph mark. We therefore conclude that H3.3 S31ph is predominantly a mitotic chromatin mark in both somatic cells and reconstituted chromatin in early embryonic extracts, a mark imposed within chromatin and not prior to deposition. Then, using *Xenopus* egg extracts and sperm chromatin, we further explored the kinetics of appearance of the modification. We detected a peak of H3.3 S31ph 30 min after metaphase entrance, similar to H3 T3ph but later in mitosis compared with H3 S10ph and H3 S28ph (Fig. 4c). Interestingly, disappearance of all phosphorylation marks showed similar kinetics after anaphase induction. In mammals, according to the current literature, kinase candidates could either be CHK1[70] or Aurora B[71]. Here, using *Xenopus* sperm chromatin in mitotic egg extracts added with various kinase and phosphatase inhibitors, we tested which kinases were critical for H3.3 S31ph in this system (Fig. 4d). Both CHK1 and Aurora B inhibitors led to a decrease of all H3 phosphorylation, including H3.3 S31ph. This suggests that both kinases are important for phosphorylation of H3.3 S31ph, possibly by impacting on each other considering the network of mitotic kinases[72,73]. In addition, we also confirmed these findings in human cells (Supplementary Fig. 4d). In contrast, inhibiting PP1/PP2 phosphatases with two different inhibitors increased all H3 phosphorylation but not H3.3 S31ph (Fig. 4d). This suggests that distinct means remove these marks. We next asked whether H3.3 S31ph was also critical at the time of gastrulation in our rescue experiments.

**A negative charge at position 31 is key during gastrulation**. To explore a potential need for the phosphorylation of H3.3 S31, we designed another H3.3 mutant carrying an aspartic acid, eH3.3 S31D. This mutant form acts as a phospho-mimic version for this residue and carries a constitutive negative charge at that cannot be dynamically regulated by kinases or phosphatases (Fig. 5a). This mutant form rescued the depletion of endogenous H3.3 to a similar extent to eH3.3 WT. Notably, all mutants were expressed and incorporated into chromatin in the same proportion (Supplementary Fig. 5a, b) and we verified that these mutations on the H3.3 tail did not indirectly alter their abilities to interact with specific histone chaperones (Fig. 5b and Supplementary Fig. 3b). In particular, neither H3.3 S31A nor S31D mutations affected the mode of incorporation of H3.3 in the *Xenopus* egg extracts–base chromatin assembly assays (Fig. 5c and Supplementary Fig. 5c). This enabled us to discard any defects that could be related to inefficient incorporation. We thus conclude that the residue at position 31 in H3.3, either as a serine or as a negatively charged

residue (phospho-mimic), is specifically needed for the function of the H3.3 variant once incorporated into chromatin as revealed at the time of gastrulation during early development in the *Xenopus*. This led us to investigate how H3.3 S31 phosphorylation impacts its binding partners and neighboring PTMs.

**H3.3 S31D can attract or repulse distinct factors**. To explore the specific roles H3.3 S31ph, we searched for specific interactors. To this end, we used biotinylated peptides corresponding to the N-terminal tail of H3.3 histones, carrying various mutations on the residue at position 31. We incubated each peptide with either interphase or mitotic *Xenopus* egg extracts for pull-down experiments and performed proteomics mass spectrometry analysis (Fig. 6a). We first ensured that all peptides were present in comparable amounts, using an antibody recognizing the N-terminal part of H3 (Supplementary Fig. 6a). In addition, in the mitotic extract, H3.3 S31 peptides showed no detectable H3.3 S31ph, as revealed with the H3.3 S31ph antibody, whereas the H3.3 S31D peptide form was readily detected, thus representing a good phospho-mimic. We identified binding partners obtained for each condition in the interphase and mitotic extracts by mass spectrometry. We compared their distribution with Venn diagrams to highlight common vs. distinct protein partners (Fig. 6b). For each specific condition, we selected the Top 5 categories according to Gene Ontology, based on *p*-values, to assess potential associated functions for factors unique to these conditions (Supplementary Fig. 6b). Furthermore, we also looked into the Gene Ontology of proteins found in common for H3.3 S31 and H3.3 S31A peptides but not retrieved in H3.3 S31D (Supplementary Fig. 6c). We focused on the H3.3 S31D peptide and displayed the Gene Ontologies of interest relative to the other peptides (Fig. 6c). Remarkably, in interphase, H3.3 S31D attracted factors involved in the β-cat-TCF pathways, namely TLE and BCL, involved in the transcriptional activation of a number of genes key for developmental programs[74]. In mitotic extracts, H3.3 S31D repulsed factors implicated in chromosome condensation, including NCAP proteins, or splicing such as SNRNP and LSM, which were enriched with both H3.3 S31 and H3.3 S31A peptides. This is particularly interesting given that previous studies found that H3.3 S31ph inhibited the binding of ZMYND11, a factor involved in intron retention that recognizes H3K36me3[75,76]. Moreover, transcription and splicing are associated with distinct PTMs, for instance, H3K27ac and H3K27me3, and H3K36me3, respectively. Thus, considering the particular attraction and repulsion properties of H3.3 S31D, it was critical to examine how this mutation could influence neighboring PTMs.

**H3.3 S31D-negative charge promotes in-*cis* H3.3K27ac in vivo**. We therefore investigated how H3.3 S31D impacted neighboring PTMs and chromatin states (Fig. 7a). For this, we first constructed new stable human cell lines expressing HA-tagged H3.3 constructs from the same genomic locus, carrying mutations at position S31 (Supplementary Fig. 7a). We verified that their cell cycle, histone chaperones, and endogenous PTMs were comparable (Supplementary Fig. 7b, c). Given the relatively low level of expression of the different exogenous H3 constructs, we enriched them by an HA pulldown, to examine the impact of H3.3 S31-negative charge on other PTMs either in-*cis* or in-*trans* (Fig. 7b). Although H3.3K27me3 was present on endogenous histones, it was undetectable on the exogenous H3.3 S31D. Moreover, H3.3K27ac was enriched on H3.3 S31D, in agreement with data obtained in mouse embryonic stem cells (mESC), although this crosstalk was observed in-*trans*[77]. We did not detect any significant changes for H3K36me3, although another phospho-mimic mutation led to a specific increase of this mark during

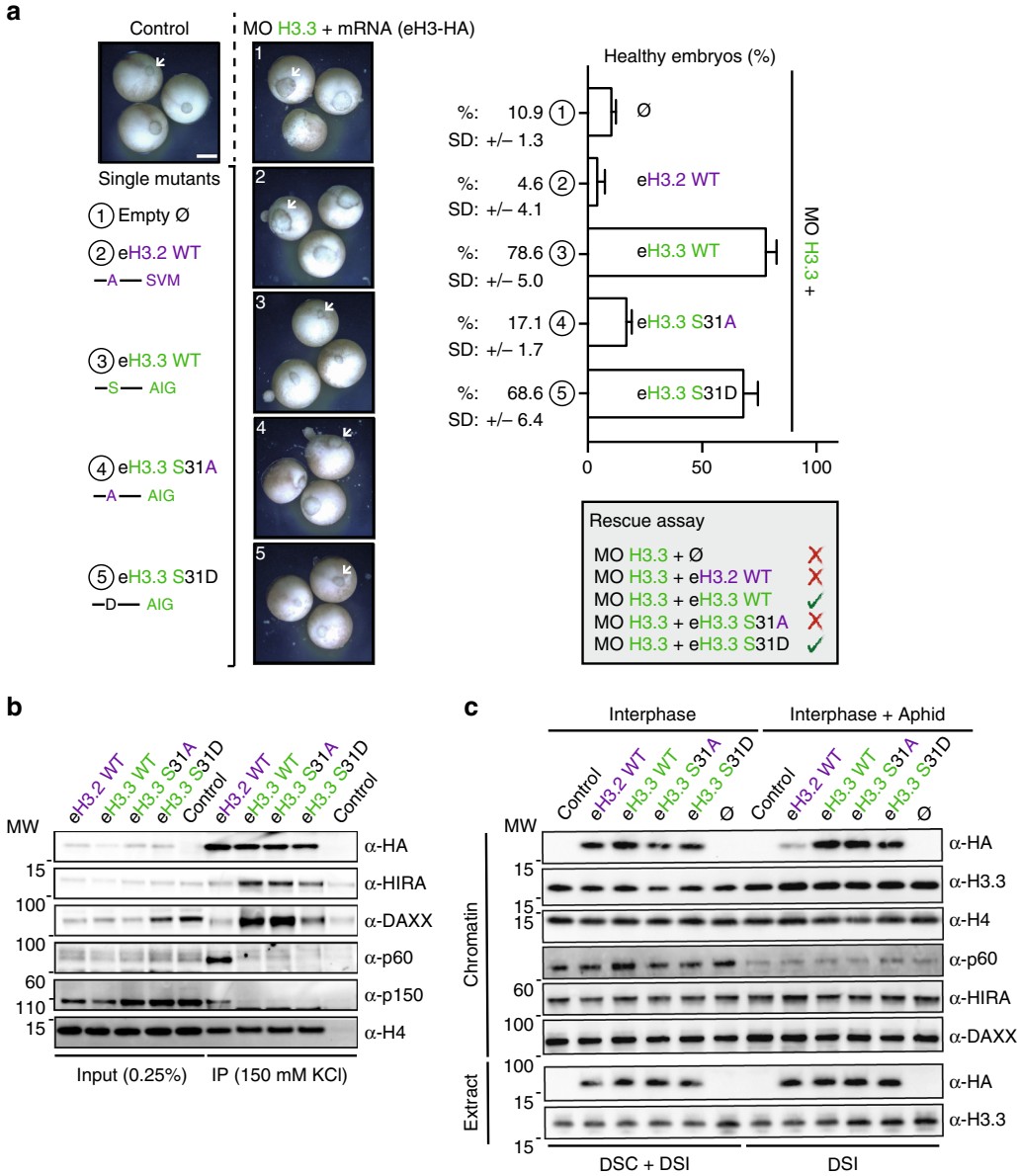

**Fig. 5 H3.3 S31D phospho-mimic rescues H3.3 depletion. a** Rescue assays with eH3.3 S31D mutant and controls. Injections are performed at two-cell stage. Effect on gastrulation is analyzed at stage 12. White arrowheads indicate the blastopore closure. Scale bar corresponds to 500 μm. Bar plot shows mean ± SD for three independent experiments with more than 30 embryos. **b** Immunoprecipitation of eH3.3 S31 mutants from interphase extracts. Recombinant proteins are produced in rabbit reticulocyte lysates and pulled down by their HA-tag after incubation. The binding of HIRA, DAXX, p60, and p150 is analyzed by western blotting with specific antibodies. α-HA antibody is used to detect eH3.3. H4 is used as an internal control. Primary antibodies indicated on the right. Molecular weight (MW) markers in kDa indicated on the left. **c** Incorporation of eH3 S31 mutant forms and controls into sperm chromatin in interphase extracts. Purified nuclei remodeled in the interphase extracts supplemented with indicated eH3.3 proteins in the presence or absence of aphidicolin are analyzed by western blotting with indicated antibodies. α-HA antibody is used to detect eH3.3. Primary antibodies indicated on the right. Molecular weight (MW) markers in kDa indicated on the left. DSC: DNA-synthesis coupled, DSI: DNA-synthesis independent mode of incorporation. Source data are provided as a Source Data file.

macrophage activation[78]. This may relate to the different cellular systems and developmental status. It was thus critical to examine which modifications could be affected in the developing *Xenopus* embryo, the system in which we established the importance of S31. We thus used embryos injected with the various HA-tagged H3.3 mutant mRNAs and carried out pulldowns at gastrulation (Fig. 7c). Remarkably, we observed a dramatic effect by pulling down eH3.3 S31D, which showed a strong signal for H3.3K27ac. Taken together, these data show a clear crosstalk between the negative charge on H3.3 S31 and neighboring PTMs, potentially

impacting transcription (Supplementary Fig. 7d). The effect on H3.3K27ac stands out as a key change in the developing embryo.

## Discussion

By exploiting depletion of endogenous H3.3 and complementation assays in *X. laevis*, and monitoring the capacity to undergo gastrulation, we disentangled the critical role of H3.3 within chromatin through amino acid 31, independently of its mode of incorporation.

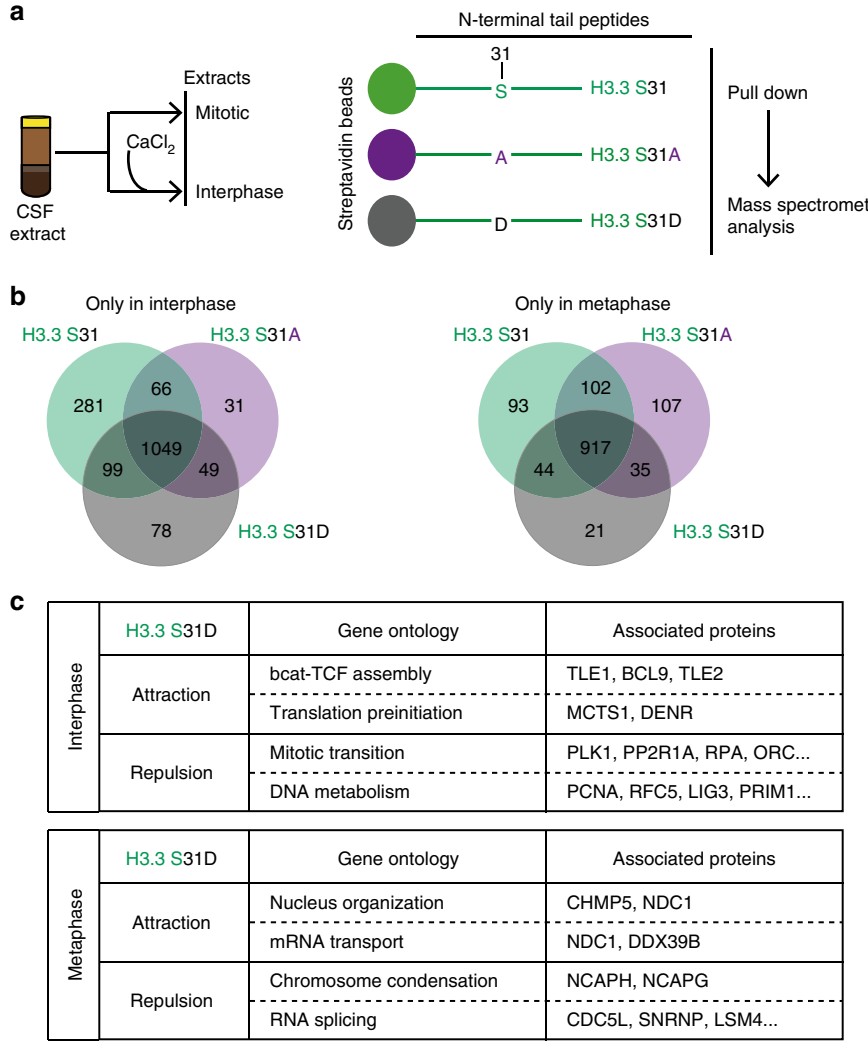

**Fig. 6 Mass spectrometry of H3.3 peptides in interphase and mitotic extracts reveals distinct protein associations. a** Peptide pull-down strategy. Biotinylated peptides corresponding to H3 N-terminal tails are incubated in mitotic and interphase *Xenopus* egg extracts before pulldown, to identify interactors by proteomics mass spectrometry analysis performed from three replicates. **b** Venn diagrams of proteins identified by mass spectrometry analyses from H3 N-terminal biotinylated peptides in *Xenopus* egg extracts. Binding partners are selected with at least three peptides in "best analysis" either in interphase or mitotic extracts, specifically. **c** Summary of Gene Ontology analysis. Attraction refers to factors pulled down with the H3.3 S31D mutant. Repulsion refers to factors that are not pulled down with the H3.3 S31D mutant. Highlights from the top *p*-value ranked Gene Ontology molecular processes for attracted and repulsed factors are displayed with a selection of associated proteins. See Supplementary Fig. 6 for all top 5 *p*-value ranked molecular processes from Gene Ontology analysis for each condition obtained with the Enrichr tool[106].

Replicative and non-replicative H3 variants show distinct genome-wide distributions[79], with enrichment of H3.3 at enhancers, or proximal to telomeres and centromeres as observed in several models in somatic cells and embryonic cells[80]. How these distinct patterns arise and evolve dynamically during development and are then maintained in given lineages remain to be established, particularly at the gastrulation stage when the different embryonic layers emerge. Moreover, how the unique properties of H3.3 influence cell cycle-related functions and/or cell-fate programming is still an open question. We first examined whether the mutations of the residues assigned to a key role in the choice of the deposition pathway[59–62] affect the capacity to rescue the depletion of endogenous H3.3. Surprisingly, an eH3.3 form containing the H3.2 recognition motif still complements H3.3 functions in our developmental assay, even though this swap between motifs effectively alters the chaperone interactions and the incorporation pathway. These data underline the fact that

neither the chaperone interaction nor the mode of incorporation of the variant is critical to enable H3.3-specific roles at this time of development in the context of endogenous H3.3 depletion. Of note, this set of data, which rely on an artificial bypass, does not undermine the importance of the loading mechanism but rather enables us to tease apart different aspects. Furthermore, the importance of distinct pathways may add to the need for distinct residues at later developmental stages beyond gastrulation. The special situation at gastrulation may relate to unique emerging functions of distinct cell fates at this time, including the progressive acquisition of chromatin properties with the addition of somatic linker histone variants[81].

Considering the transient nature of the morpholino depletion, other approaches would be necessary to access later stages. Most importantly, for *Xenopus* early development, and in sharp contrast with what one would have anticipated, it is the presence per se of H3.3 into chromatin that proves most important in vivo,

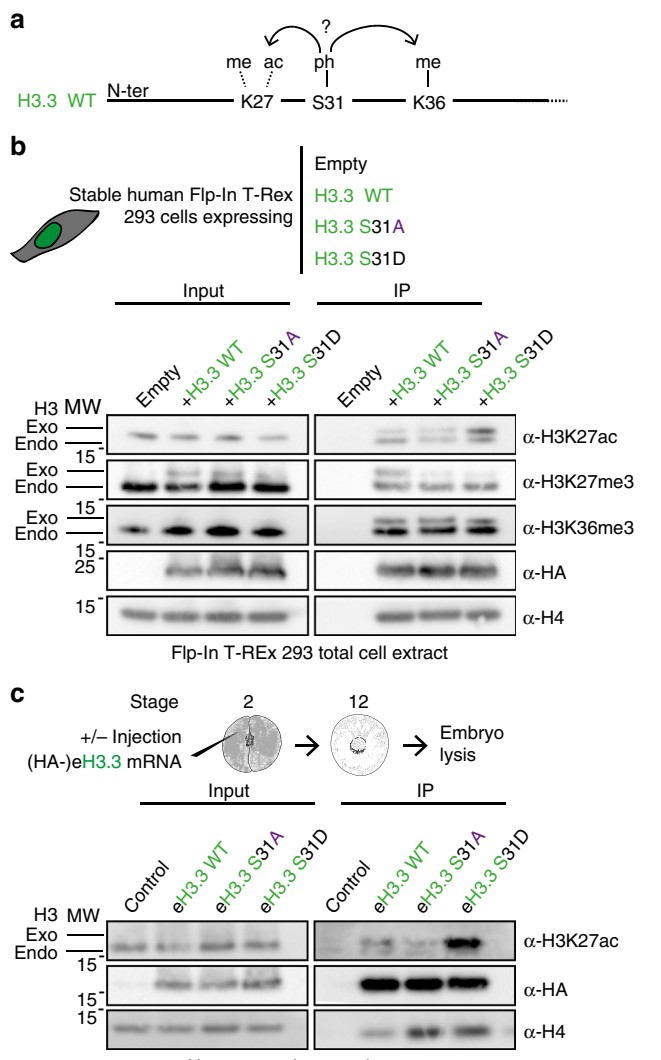

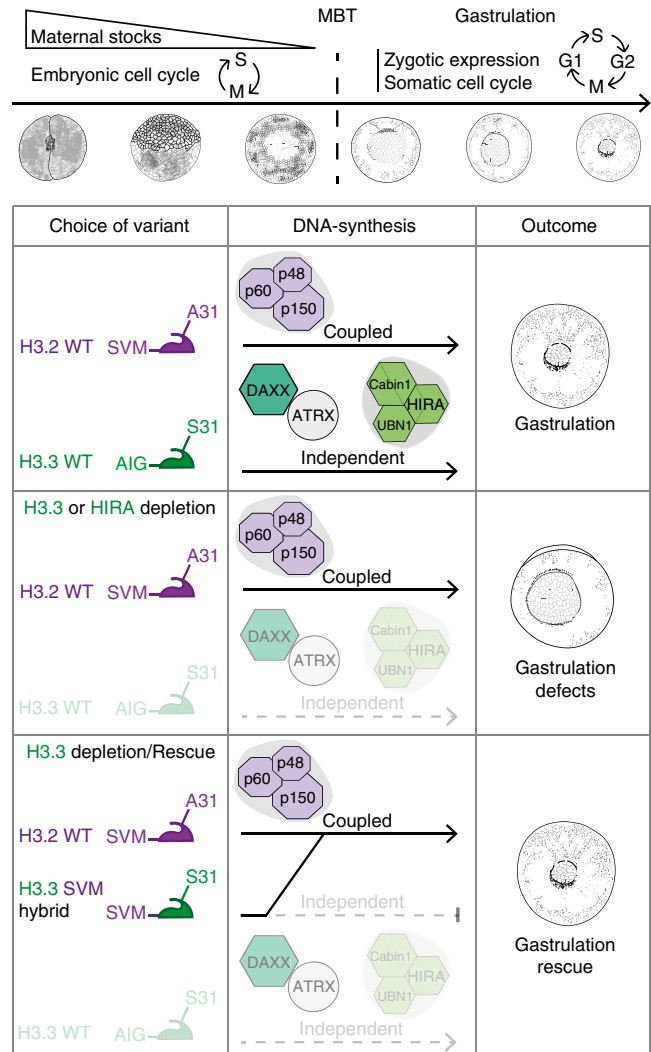

**Fig. 7 H3.3 S31D promotes H3K27ac and reduces H3K27me3 in-*cis*.**
**a** Scheme of potential crosstalk of H3.3 S31ph with neighboring PTMs. H3.3 S31 is located between H3.3K27 and H3.3K36, which can both be modified. **b** PTM crosstalk analyzed in H3.3-transfected Flp-In T-Rex 293 cell lines. Immunoprecipitations of H3.3 mutants enabled to enrich for the exogenous form. Presence of PTMs is analyzed by western blotting with indicated antibodies. α-HA antibody is used to detect eH3.3. H4 is used as an internal control. Primary antibodies indicated on the right. Molecular weight (MW) markers in kDa indicated on the left. Exo: exogenous H3.3, endo: endogenous H3 and H3.3. **c** PTM crosstalk in *Xenopus* embryos. Immunoprecipitations of eH3.3 mutants and controls are performed at stage 12 after their injection at the 2-cell stage and are analyzed by western blotting with indicated antibodies. α-HA antibody is used to detect eH3.3. H4 is used as an internal control. Primary antibodies indicated on the right. Molecular weight (MW) markers in kDa indicated on the left. Exo: exogenous H3.3, endo: endogenous H3 and H3.3. Source data are provided as a Source Data file.

**Fig. 8 Model.** Defects associated with H3.3 depletion can be rescued with H3 histone variants carrying a potential negatively charged residue, regardless of the mode of incorporation. MBT: midblastula transition.

regardless of the means of incorporation (Fig. 8). This discovery shed light on unique roles performed by the single amino acid S31 on H3.3, highlighting the idea that every amino acid matters[8]!

By replacing the serine at position 31 in the N-terminal tail of H3.3 with the alanine found in H3.2, we showed that the eH3.3 S31A mutant could not replace H3.3. This demonstrates that H3.3 S31 is essential in our complementation assay after endogenous H3.3 depletion. This serine on H3 can undergo

phosphorylation, whereas the alanine on H3.2 cannot. This specific phosphorylation of H3.3 shows enrichment during metaphase in human cell line[63]. In our system, we revealed a similar dynamics in a *Xenopus* cell line. Although only phosphorylated within chromatin, H3.3 S31ph peaks in mitosis later than H3 S10ph and H3 S28ph. Although other studies have looked for additional kinases involved in H3.3 S31ph during interphase, including IKKα[82], either CHK1[70] or Aurora B[71] has been previously identified for mitosis. Here we found that both CHK1 and Aurora B contribute in mitosis to H3S10 and H3.3 S31 phosphorylation, and we identify this modification as a late event for H3.3 S31. Here we show that the dynamics of this mark is not required for proper gastrulation. Indeed, a phospho-mimic form with a constitutive negative charge on the serine at position 31, H3.3 S31D, could readily rescue the depletion of H3.3. We can foresee several possible explanations. On the one hand, the negative charge at position 31 could be an absolute requirement for H3.3 functions regardless of any dynamics. On the other hand, the negative charge on this residue could be associated with mechanisms that occur solely in mitosis and the dynamics may simply be ensured by removal of the variant without invoking a particular phosphatase. Interestingly, although inhibition of PP1/ PP2 phosphatases increased H3 T3ph, H3 S10ph, and H3 S28ph

levels, it does not affect H3.3 S31ph. This is consistent with our hypothesis of H3.3 S31ph removal by eviction but also brings an intriguing possibility for the mark to be maintained through mitosis at certain loci, where it could act as a bookmarking. Future work should explore these possibilities.

The exact role of H3.3 S31 remains enigmatic. Although we and others[63–65] detect strong H3.3 S31ph signals primarily in mitosis, we cannot exclude that discrete genomic sites are marked in interphase but were not detected in our assays. Indeed, this mark has been associated with particular transcribed regions in activated macrophages[78,82] and mESC enhancers[77]. Our assays demonstrate the importance of H3.3 S31 at gastrulation, a critical time for lineage commitment accompanied by dramatic cell cycle changes and modifications in gene expression patterns. Taking into account the specific importance of H3.3 that ranges from transcription to reprogramming[20,83,84], the observed developmental defects could arise from a role for H3.3 S31 and its phosphorylation in transcription initiation or maintenance. Indeed, we could detect an increase in-cis of H3.3K27ac with the H3.3 S31D construct, along with a decrease of H3.3K27me3. These marks are well-characterized for their involvement in transcription and this is in line with previous observations showing that H3.3 depletion leads to transcriptional defects[58]. It will be critical to investigate in more details the nature of the transcripts affected when rescuing with the phospho-mimic version. Furthermore, and similar to the strategy used for BAP1 depletion[85], we envisage the use of histone deacetylases inhibitors during development when endogenous H3.3 is depleted to potentially overcome gastrulation defects by restoring H3K27ac levels. Importantly, MBT in *Xenopus* also leads to changes in cell cycle progression, including establishment of longer cell cycles (somatic type) with gap phases and acquisition of checkpoints[55,56]. Histones play a key role in regulating the start of the MBT, possibly through titration mechanisms[86,87]. It would therefore be of interest to explore the role of specific variants in this context, specially in the case of acetylation and histone variant crosstalk[88]. In addition, previous studies have suggested that the presence of H3.3 could affect chromosome segregation. Indeed, in double H3.3 knockout mESCs, an increase in anaphase bridges and lagging chromosomes has been observed[89]. Furthermore, H3.3 S31ph can coat lagging chromosomes, which is associated with p53 activation and the prevention of aneuploidy[65]. Considering H3.3 S31ph accumulation at centromeres in mitosis[63], it is tempting to speculate that H3.3 S31ph has a crosstalk with CENP-A incorporation at late mitosis and the beginning of G1. Interestingly, CENP-A incorporation at the centromere has been shown to be dependent on both the presence of H3.3 as a placeholder[90] and on transcription[91,92]. To reconcile both aspects, we envision that H3.3 S31ph could act as a phospho-switch in mitosis to regulate transcription at critical chromosomal landmarks and in interphase to control and maintain a transcription program. Alternatively, one could also consider that the alanine present on H3.2 could prevent particular interactions specific to H3.3 S31 or H3.3 S31ph, and permitted by an aspartic acid. Notably, in *Arabidopsis*, the replicative form cannot be rescued by H3.3, or more specifically the A31T mutation[93]. Indeed, ATXR5, a plant-specific H3K27 methyltransferase, specifically recognizes H3.1 and not H3.3[94]. Therefore, the alanine at position 31 on replicative histone H3.1 prevents the heterochromatinization of H3.3-rich regions during replication. Thus, in plants, the replicative histone H3 variant avoids the presence of a negative charge at position 31. It also suggests that the specific residue at position 31 of the replicative and non-replicative H3 variants may exhibit different functions by promoting or excluding specific binding partners. Therefore, the mass spectrometry data we have generated in our *Xenopus*

studies provides an important source of information to explore further. The fact that H3.3 S31D can attract factors in interphase involved in the β-catenin pathway or repulse proteins in metaphase involved in chromosome condensation and mRNA splicing are first insights, but we believe that there will be more to discover. For instance, H3.3 S31D may impeach on specific developmental programs, as the β-catenin pathway is important in the epithelial–mesenchymal transition that occurs following gastrulation and is also implicated in metastatic signaling in cancer[74,95]. Also, the impact of H3.3 S31D on the mitotic condensing machinery will be another challenge to consider for the future, opening new avenues on the control of mitosis and chromosome organization.

As H3.3 S31-negative charge has been linked to intron retention and pre-mRNA processing by preventing binding of the tumor suppressor ZMYND11[75,76], further studies will be essential to better delineate the role of H3.3 S31ph in these contexts and the different proteins involved. The presence of H3.3 S31D also alters neighboring residues in-cis. Conversely, other modifications on the H3 tail could themselves impact H3.3 S31 modification and in this respect, the effects of mutations affecting neighboring residues will be interesting to explore. Notably, as H3.3 S31 is close to residues often mutated in aggressive cancers such as H3 K27M and G34R in pediatric glioblastoma[42–44,47,96], it will be interesting to evaluate the impact of these onco-histone mutations on H3.3-specific phosphorylation as well. Altogether, we show that H3.3 S31 is the key residue that confers specific functions to H3.3 within chromatin, compared with its H3.2 counterpart. It also establishes the importance of a distinct histone variant residue for the proper development of a vertebrate during gastrulation. Future work should explore whether a similar requirement also occurs in mammals to provide a comprehensive view of the importance of the non-replicative variant H3.3 and its role during vertebrate development and in disease states.

## Methods

**X. laevis embryo manipulation.** We used *X. laevis* adults (2 years old) from the Centre de Ressource Biologie Xenope. We prepared embryos at 18 °C as in ref. [88] and staged them according to ref. [53]. We acquired embryos during gastrulation with a MZFLIII magnifier (Leica) and the SPOT software. Animal care and use for this study were performed in accordance with the recommendations of the European Community (2010/63/UE) for the care and use of laboratory animals. Experimental procedures were specifically approved by the ethics committee of the Institut Curie CEEA-IC #118 (Authorization APAFIS#11226-2017091116031353-v1 given by National Authority) in compliance with the international guidelines. D. S., E.B., and G.A. possess the authorization for vertebrates' experimental use.

**Plasmid cloning and mRNA transcription.** We cloned all H3 cDNAs in the pβRN3P vector[97]. This vector stabilizes RNA and improves their translation efficiency, while injected into *Xenopus* eggs. In addition, an HA-tag has been inserted in the C-terminal of H3. We obtained mRNAs by in vitro transcription of PCR-amplified fragments of the different pβRN3P vectors (forward: 5′-gtaaaacgacggcc agt-3′ and reverse: 5′-ggaaacagctatgaccatga-3′). We transcribed mRNAs starting with 5 ng of PCR-amplified fragment, 10 µL Buffer 5×, 5 µL of dithiothreitol (DTT) 100 mM, 0.25 µL of bovine serum albumin (BSA) 10 mg/mL (NEB), 5 µL of ATP, CTP, UTP 10 mM, 1.65 µL of GTP 10 mM (Sigma-Aldrich), 3.35 µL of Me7GTP 10 mM (NEB), 2 µL of RNasin Plus RNase Inhibitor (Promega), and 50 µL H2O qsp. After 10 min on ice, we added 2 µL of T3 RNA Polymerase (Promega) to each sample, incubated for 30 min at 37 °C, then added 0.5 µL of fresh T3 RNA Polymerase and incubated for another 10 min at 37 °C. After DNA digestion with 2 µL RQ1 DNase (Promega) for 20 min at 37 °C, we extracted mRNAs with phenol–chloroform and purified them through Sephadex G-50 Quick Spin Column for radiolabeled RNA purification (Sigma-Aldrich), previously equilibrated six times with 1 mL of TE 10 mM.

**Morpholino and mRNA microinjection into X. laevis embryos.** We microinjected two-cell embryos using a Brinkmann micromanipulator and a Drumond microinjector on two-cell stage eggs with an injection volume set to 9.2 nL, to deliver the appropriate quantity of morpholino and mRNAs (1× = 4.6 ng). Morpholino and mRNA concentrations have been optimized for efficient depletion and rescue at gastrulation. More than 30 embryos are injected per

condition, with at least 3 biological replicates. We knocked down endogenous H3.3 with morpholino designed to bind to the initiation region of *X. laevis* H3.3 mRNAs from gene b as described in ref. [58]. We selected this gene given its peak of expression specifically at gastrulation[98]. The morpholino only targets endogenous H3.3, as verified in details[98]. H3.3 morpholino sequence 5′-GGTCTGCTTTG TACGGGCCATTTCC-3′ targets 5′-GGAAATGGCCCGTACAAAGCAGA CC-3′. *Xenopus* H3.3 sequence for the gene b is 5′-$_{ATG}$GCCCGTACAAAGCAG ACTGCCCGTAAA-3′, whereas *Xenopus* H3 sequence is 5′-$_{ATG}$GC**T**CGTAC**T**A AGCAGACCGCTCGCAAG-3′. eH3 sequences used in the study start as 5′-$_{ATG}$GCCCG**A**AC**C**AAGCAGACTGCTCGTAAG-3′.

$_{ATG}$ corresponds to the first codon and mismatch DNA bases are in bold and underlined.

**X. laevis embryo protein extract and western blotting**. We prepared total protein extracts from *X. laevis* embryos using the CelLytic Express reagent (Sigma-Aldrich) and centrifuging at 20,000 × *g* for 20 min at 4 °C the extracts after 30 min incubation on ice. We analyzed protein samples by electrophoresis on 4–12% NuPAGE SDS-polyacrylamide electrophoresis gels with MES SDS Running buffer (Life Technologies) and corresponding LDS buffer NuPage (Invitrogen) with DTT. Primary antibodies were detected using horseradish peroxidase-conjugated secondary antibodies (Jackson Immunoresearch Laboratories) and SuperSignal West Dura Extended Duration Substrate (ThermoFisher). The signal was acquired using ChemiDoc Imager (Biorad).

**X. laevis embryo fractionation**. We collected 30 embryos at stage 12 and performed fractionation as in ref. [58]. We lysed the embryos in 200 μL of Lysis Buffer 1 (15 mM Tris-HCl pH 7.5, 300 mM NaCl, 5 mM MgCl$_2$, 10 mM β-glycerophosphate, and protease inhibitors) and centrifuged them at 1000 × *g* for 4 min at 4 °C. We collected supernatant (soluble fraction) and cleared it with ultracentrifugation at 210,000 × *g* for 30 min at 4 °C. We washed the chromatin fraction three times with Lysis buffer and once with Buffer A (15 mM Tris-HCl pH 7.5, 15 mM NaCl, 60 mM KCl, 0.34 M sucrose, 1 mM DTT, 10 mM β-glycerophosphate, and protease inhibitors). We performed MNase (Sigma) digestion during 12 min with 2.5 U/mL final concentration, after addition of CaCl$_2$ (1 mM final). We stopped the reaction by EDTA (4 mM final) and recovered solubilized chromatin fraction after a ultracentrifugation at 210,000 × *g* for 30 min at 4 °C.

**X. laevis embryo immunoprecipitation**. We collected embryos at stage 12 and prepared either total embryo extracts or soluble fractions as above. We ultra-centrifuged total extracts at 210,000 × *g* for 30 min at 4 °C. We used 100 μg of protein for each condition for immunoprecipitation with anti-HA magnetic beads (Thermo Scientific) overnight at 4 °C in 400 μL of IP buffer (20 mM Tris-HCl pH 7.5, 15 mM KCl, 150 mM NaCl, 10% glycerol, 0.1 mM EDTA, 10 mM β-glycer-ophosphate, 0.01% NP-40, 1 mM DTT, and protease inhibitors). After three washes, we eluted proteins with LDS buffer NuPage (Invitrogen) with the reducing agent and analyzed samples by electrophoresis as described above.

**X. laevis egg extract preparation**. We prepared *X. laevis* sperm nuclei and low-speed extracts arrested by cytostatic factor (CSF) of *X. laevis* eggs as previously described[99]. Briefly, we collected eggs freshly and centrifuged them at low speed (16,000 × *g*) to conserve the mitotic phase, below lipids. We induced interphase by the addition of CaCl$_2$ at the final concentration 0.06 mM to CSF-arrested egg extracts. We added sperm chromatin at a concentration 1000–4000 nuclei/μl. After DNA replication, a 2/3 volume of the CSF-arrested extract was added to induce mitosis. For experiments in Fig. 4c, a second addition of CaCl$_2$ at a final concentration of 0.06 mM induced transition to the second interphase. For the experiments of chromatin assembly in the presence of kinase or phosphatase inhibitors, we supplemented CSF-arrested extracts with the indicated concentration of Hesperadin hydrochloride (#3988, Tocris), ZM 447439 (#2458, Tocris), SB 218078 (#2560, Tocris), PF 477736 (#4277, Tocris), and Calyculin A (#208851, Calbiochem) prior to the addition of 5000 sperm nuclei/μl. Incubation lasted for 50 min at room temperature. For treatment with okadaic acid, similarly using sperm nuclei, we first allowed chromatin assembly in CSF-arrested extracts treated with 20 μg/ml nocodazole (Sigma) for 30 min at room temperature. Next, we treated the reaction with either dimethyl sulfoxide or 1 μM okadaic acid (#459620, Calbiochem) for another 20 min. We isolated the assembled chromatin and analyzed by western blotting as described above.

**X. laevis egg extract immunoprecipitation**. Briefly, we produced recombinant H3 mutant proteins from mRNAs using rabbit reticulocyte lysate (Promega L4600). After 3 h of incubation at 4 °C in interphase extracts followed by another 3 h incubation with anti-HA beads, we pulled down and washed the proteins in 0.8× CSF-XB buffer (10 mM Hepes-KOH pH 7.7, 100 mM KCl, 2 mM MgCl$_2$, 5 mM EGTA) containing 5% glycerol, 0.5% Triton X-100, and protease and phosphatase inhibitors. We eluted proteins with LDS buffer NuPage (Invitrogen) with the reducing agent and analyzed samples by electrophoresis as described above.

**Histone deposition assays**. We added 100,000 sperm nuclei to each 150 μL of corresponding extracts with or without aphidicolin (50 μg/mL). We then supplied 15 μL of the rabbit reticulocyte lysate used to produce recombinant H3 mutant proteins from mRNAs. After 40 min of incubation at room temperature, we purified chromatin and we analyzed protein samples by electrophoresis as described above.

**X. laevis sperm chromatin purification**. We diluted fivefold 100 μl aliquots of each reaction with 0.8× CSF-XB buffer containing 20 mM β-glycerophosphate, 5% glycerol, and 0.5% Triton X-100, which we incubated for 1 min at room temperature. We then layered the samples onto a 35% glycerol-containing CSF-XB cushion and centrifuged them at 10,000 × *g* for 5 min at 4 °C. We resuspended the pellets in the same buffer and repeated the centrifugation step. For purification of interphase chromatin, we diluted 100 μl aliquots of extract with 0.8× CSF-XB buffer containing 20 mM β-glycerophosphate and 5% glycerol, which we incubated for 1 min at room temperature, followed by centrifugation through the cushion at 10,000 × *g* for 5 min at 4 °C. We resuspended purified chromatin directly in LDS buffer NuPage (Invitrogen) with NuPage reducing agent (Invitrogen). We analyzed samples by electrophoresis as described above.

**WebLogo sequence alignment**. After performing multiple sequence alignment using MUSCLE[100], we displayed the alignments using WebLogo3[101] with probability units.

**Antibodies**. See Supplementary Data 1 and Supplementary Fig. 8.

**Generation of Flp-In T-Rex-293 cell lines**. We have inserted fragments corresponding to HA-tagged *Xenopus* H3.3 WT, H3.3 S31A, and H3.3 S31D cDNAs into pcDNA5/FRT/TO (Invitrogen) using BamHI/NotI restriction enzymes. We co-transfected these plasmids into Flp-In T-Rex 293 cells (Invitrogen), together with Flp-recombinase expression vector pOG44 (Invitrogen) using JetPRIME (Polyplus). We selected stably transfected cells with 150 mg/mL Hygromycin B (Gibco). We induced histone expression by adding 1 μg/ml doxycycline at least 120 h before analysis. For total-extract western blotting analysis, an equal number of cells were resuspended directly in LDS buffer NuPage (Invitrogen) with NuPage reducing agent (Invitrogen) and Universal Nuclease (Pierce).

**Total cell extracts preparation and immunoprecipitation**. We resuspended dry cell pellets in an equal volume of the lysis buffer (50 mM Tri-HCl pH 7.5, 300 mM NaCl, 0.5% NP-40, 10% glycerol, 2 mM MgCl$_2$, 10 μM CaCl$_2$, 5 mM EGTA-KOH pH 8.0, 1 mM DTT, and protease inhibitors) and incubated these suspensions for 30 min at 4 °C. We then supplemented these samples with CaCl$_2$ up to 1 mM and treated with 2.5 U/ml MNase (Sigma) for 12 min at 37 °C. The reaction was stopped by addition of EDTA-NaOH, pH 8.0 up to 4 mM. We ultracentrifuged the extracts at 100,000 × *g* for 30 min and diluted supernatant with an equal volume of dilution buffer (50 mM Tris-HCl pH 7.5, 10% glycerol, 100 mM KCl, 5 mM EGTA-KOH pH 8.0, 1 mM DTT, and protease inhibitors). We used 500 μg total protein for IP with anti-HA magnetic beads (Thermo Scientific) for an overnight incubation at 4 °C in 500 μL of IP buffer (20 mM Tris-HCl pH 7.5, 15 mM KCl, 150 mM KCl, 10% glycerol, 0.1 mM EDTA, 10 mM β-glycerophosphate, 0.01% NP-40, 1 mM DTT, and protease inhibitors). After three washes with IP buffer, we eluted proteins with LDS buffer NuPage (Invitrogen) and reducing agent (Invitrogen) to process samples for electrophoresis analysis as described above.

**Fluorescence-activated cell sorting**. Flp-In T-Rex 293 cells were washed twice in phosphate-buffered saline (PBS) and trypsinized. They were recovered in PBS and centrifuged at 300 × *g* for 5 min. Cell pellets were then resuspended in 500 μL of PBS and dropped in 1 mL of cold ethanol, while vortexing it. Cells were fixed at 4 °C for at least 30 min. After a centrifugation at 300 × *g* and a wash of PBS, cell pellets were resuspended in 500 μL of FxCycle PI/RNase Staining Solution (ThermoFisher Scientific) and further incubated at 37 °C for 30 min. Cells were finally filtered with Cell Strainer Snap Caps (Corning) and processed for analysis on an Accuri C6 flow cytometer (BD). Data were analyzed using FlowJo.

**Proteomics and mass spectrometry analysis**. Streptavidin magnetic beads (30 μL; Thermo Scientific) were washed with PBS and incubated with 60 pmol of the different peptides (GeneCust) corresponding to H3 N-terminal tails (H3.3 WT, H3.3 S31A, and H3.3 S31D) in 300 μL of PBS. After 2 h of incubation and a 30 min block with BSA at room temperature, beads coupled to peptides were incubated 3 h with either mitotic or interphase eggs extracts diluted 5× in 400 μL total CSF-XB buffer, supplemented with 10% glycerol, 0.1% Triton, 0.1% Tween-20, and 1 mM DTT. Beads coupled to peptides were finally washed three times in PBS before digestion. Proteins on magnetic beads were washed two additional times with 100 μL of 25 mM NH$_4$HCO$_3$, to eliminate the remaining detergents. Beads were resuspended and digested by adding 0.2 μg of trypsine/LysC (Promega) in 100 μL of 25 mM NH$_4$HCO$_3$ for 1 h at 37 °C. Peptides were desalted and concentrated using homemade C18 StageTips. After elution, peptides were analyzed using an RSLCnano system (Ultimate 3000, Thermo Scientific) coupled online to an

Orbitrap Fusion Tribrid mass spectrometer (Thermo Scientific) as in ref. [102]. For identification, the data were searched against the *X. laevis* Database (June 2019) from Xenbase.org and a database containing the common contaminants using Sequest[HF] through proteome discoverer (version 2.2). Enzyme specificity was set to trypsin and a maximum of two-missed cleavage site have been allowed. Oxidized methionine, N-terminal acetylation, and carbamidomethyl cysteine were set as variable modifications. Maximum allowed mass deviation was set to 10 p.p.m. for monoisotopic precursor ions and 0.6 Da for tandem mass spetrometry peaks. The resulting files were further processed using myProMS v3.6[103] (Supplementary Data 2). False discovery rate calculation used Percolator and was set to 1% at the peptide level for the whole study. Specific proteins, selected based on at least three peptides in the best analysis (replicates $n = 3$), were analyzed further.

**Immunofluorescence and epifluorescence microscopy**. We fixed A6 cells on coverslips for 20 min in 4% paraformaldehyde, Flp-In T-Rex 293 and HeLa B cells in 2% paraformaldehyde before permeabilization with 0.2% Triton X-100. We blocked them for 45 min with 5% BSA. We then incubated coverslips with primary and secondary antibodies, and stained them with 4′,6-diamidino-2-phenylindole. We mounted the coverslips in Vectashield medium. We used a Confocal Zeiss LSM780 and we acquired images using 63×/1.4 numerical aperture under Zen blue software (Zeiss Germany) and analyzed the data using ImageJ.

**Reporting summary**. Further information on research design is available in the Nature Research Reporting Summary linked to this article.

## Data availability
The data that support this study are available from the corresponding author upon reasonable request. The mass spectrometry proteomics data have been deposited to the ProteomeXchange Consortium via the PRIDE partner repository[104] with the dataset identifier PXD016497. Source data are provided as a source data file.

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

## Acknowledgements

We thank Nezha Benabdallah for preliminary work. We also thank Dominique Ray-Gallet, Daniel Jeffery, and Jean-Pierre Quivy for critical reading, and members of UMR3664 for helpful discussions. We thank Mary Dasso and Aaron Straight for xEmi2 and xDAXX, and xCENPA antibodies, respectively, and Valentin Sabatet for mass spectrometry analysis. We acknowledge the Institut Curie PICT-IBIsA@Pasteur. La Ligue Nationale contre le Cancer (Equipe labellisée Ligue), ANR-11-LABX-0044_DEEP and ANR-10-IDEX-0001-02 PSL, ERC-2015-ADG- 694694 "ChromADICT," and ANR-16-CE12-0024 "CHIFT" support the Almouzni laboratory. The Mass Spectrometry and Proteomics Facility benefits from "Région Ile-de-France" and Fondation pour la Recherche Médicale grants. PSL university funded D.S., Ph.D., with a MESRI fellowship.

## Author contributions

G.A., D.S., and E.B. conceived the overall strategy and wrote the paper. G.A. supervised the work. D.S. performed *X. laevis* cellular and mass spectrometry experiments, and analyzed data. E.B. performed *X. laevis* egg extract and cellular experiments, and analyzed data. F.D. carried out the mass spectrometry experimental work under the supervision of D.L. Critical reading and discussion of data involved D.S., E.B., and G.A.

## Competing interests

The authors declare no competing interests.
