## [Peer Review File · Nature Communications]

Reviewers' comments:

Reviewer #1 (Remarks to the Author):

The manuscript "Regardless of the deposition pathway, amino acid 31 in histone variant H3 is essential at gastrulation in *Xenopus*" by Sibon et al. is an insightful study investigating functional consequences in *Xenopus* development by mutating H3.3 specific residues. The main and surprising discovery is the observation that amino acids required for H3.3-specific chaperone binding and subsequent variant chromatin deposition are not required for embryonic frog development. Yet, the unique serine 31 that has been found to be phosphorylated during mitosis is essential, most likely in a phosphorylated, negatively charged manner. These findings are important and novel, unfortunately, experiments investigating the mechanism by which amino acid 31 contributes to proper embryonic development are missing. It is very nice descriptive study, but at least one experimental hint towards the mechanism should be provided. My major and minor concerns are listed in detail below:

Major concerns:

- The authors state that an H3F3B morpholino was used. Did the authors also test an H3F3A morpholino and do they know anything about the expression of both H3.3 genes during *Xenopus* development? Did the author test and verify H3.3 knockdown efficiency with immunoblots?
- Is the same phenotype observed in other *Xenopus* species?
- Do the authors find more apoptotic cells in their H3.3 depleted embryos? The movies somehow suggest this.
- Figure 2B: The quantification of the rescue with WT H3.3 is not shown. As the rescue seems to reach around 70% (the scaling is not detailed enough), it would be good to compare this number to the WT control. The rescue seems high, but it is not clear whether the single chaperone binding site mutants are as good as WT H3.3 in rescuing the phenotype in this particular experiment. Additionally, have the authors looked at later stages? Is the development to a tadpole normal or are there problems?
- Figure 4A: The authors did not test whether eH3.3 SVM is still able to bind DAXX/ATRX. Most likely not, but it should be tested, as it might explain the rescue ability of this mutant.
- Figure 4: It would be important to perform ChIP-seq (or ChIP-qPCR in case specific sites are already known) with the different mutants to show that they are deposited into different regions than WT H3.3 (including heterochromatin regions) and are still able to rescue the morpholino phenotype. Just the demonstration of generalized chromatin deposition is not sufficient.
- If technically possible, the authors should provide anti-H3.3S31ph stainings of different embryonal developmental stages to determine the exact appearance and tissue localization of this modification. Is H3.3 S31 phosphorylated around blastopore stage, explaining the observed lethality of S31A mutants at this particular time during development?
- The main and most important question is about the function of H3.3S31 (and its phosphorylation) during development. Why is it so important regardless of its site of deposition? I am missing experiments trying to decipher the mechanism behind H3.3S31ph function in development. The authors raise several interesting ideas in the discussion section (role of H3.3S31ph during cell cycle, in genome stability, for CENP-A incorporation, for transcription regulation,...), but do not experimentally address them. It should be possible to look at cell cycle distributions of mutant embryos, the ploidy and whether CENP-A is incorporated at centromeres. Importantly, the authors should look into transcriptome changes and whether those might provide a clue into the process that went wrong in S31A mutants.

Minor:

- The authors talk about „DNA synthesis coupled (DSC)“ or „DNA synthesis independent (DSI)“ mechanisms. In the large majority of papers on histone variants the phrases „replication dependent RD)“ or „replication independent (RI)“ have been established. It would be preferable to switch back to RD or RI throughout the text.

- Figure 1A and B are largely redundant and have been depicted in many other publications. I suggest to move one of these Figures into the Supplement.
- I am missing molecular weight indications next to immunoblots.
- Legends for Figures 2B, 3, 5A, and 6A: Indicate numbers of embryos analyzed and whether error bars are calculated as SD or SEM.
- Supplemental Figures 2B and 3B: Control antibodies to show efficient and specific fractionation are missing. (remove one "n" from "fractionnation" in the figure).
- Nomenclature: please use the term "H3.3S31ph" instead of "H3.3S31p". It looks to similar to a mutation to a proline.

Reviewer #2 (Remarks to the Author):

In this manuscript, Sitbon et al test the *Xenopus laevis* developmental function of histone H3.3 variant amino acids. Other than altered chaperone binding and deposition timing, the functional significance of the H3.3 amino acid differences from deposition-dependent H3.2 has been enigmatic in all model systems probed. Therefore, this study, employing both *Xenopus* embryos, cell-free extracts, and cultured cells, is informative. Overall, this is a well-executed and concise study, generally supporting the conclusion that S31 phosphorylation is essential for *Xenopus laevis* gastrulation. Additionally, the authors show clear evidence that altering the chaperone-mediated deposition pathway does not influence viability (which is rather remarkable, and contrary to much of the collected wisdom of the last 10 years or so on these variants). The authors identify a striking role for acidic character at residue H3.3S31 that does not need to be dephosphorylated for the embryo to proceed through gastrulation. However, the authors do not test what the role of this phosphorylation may be, nor do they propose any experiments to decipher this question. While this work should be published, I strongly recommend that the authors consider additional experiments to answer what biological or biochemical function H3.3S31ph is performing. I have a few general and specific comments.

No specific evidence is shown of H3.3 morpholino knockdown, only the rescues are shown. In particular, the morpholino titration (Fig S1C) would be more compelling—and rule out off target effects—with direct evidence of corresponding loss of H3.3 protein. As the authors have a general H3 and an H3.3 specific antibody, this should be straightforward (as done either by TAU or western blot in the prior Almouzni lab paper Szenker et al 2012).

In Supplemental Fig 2, it appears that there may be substantial soluble (nuclear or cytoplasmic?) H3, but not soluble H4. This is curious, especially at Stage 12 embryos. Are these cytoplasmic histones bound to chaperones? As both the endogenous (Control) and rescued embryos contain soluble histone (and HA tagged histone), this suggests a bona fide biological role. Titrating the dose of rescue mRNA could address this question.

In the cell-free extract experiments, the authors should also blot for H4 (and either H2A or H2B as a negative control). H4 enrichment in the tagged H3.2 and H3.3 will ensure that it is truly binding to chaperones as a native dimer or tetramer complex. In Fig 4B, the aphidicolin experiment is cleverly designed to demonstrate replication-independent deposition.

Since the modest prior literature on H3.3S31ph and Fig 5 shows that it has different kinetics than H3S10ph (e.g. it appears later in metaphase), I recommend using cycling egg extract for a fine-grained comparison of the two marks. In particular, the extract can be supplemented with a high concentration of purified recombinant H3.3S31A and H3.3S31D (in complex with H4) to ask where in mitosis the mutation elicits a defect. This may permit insight into the specific mitotic timing of the gastrulation defect (better than the correlation with the analysis of A6 cells).

Similarly, identification of the kinase responsible for the phosphorylation in *Xenopus* would help

understand the function. Using either a Chk1 or Aurora B inhibitor in the extract experiments would help answer this question, and permit more functional understanding.

Reviewer #3 (Remarks to the Author):

The work by Sitbon et al investigates the role of the histone H3.3 variant in regulating developmental mechanisms in *Xenopus*. Incorporation of H3.3 is emerging as fundamental mechanism in chromatin remodelling, and genetic alterations affecting H3.3 and/or its deposition machinery have been reported in multiple human cancers. It is therefore important to dissect both function and regulation of this variant as this could provide key insights into pathogenesis of several neoplasms. In this manuscript, the authors report that the ability of H3.3 to regulate development is independent from its loading mechanisms, as mutating the chaperone-interacting motif of H3.3 to the one carried by the S-phase restricted H3.2 variant does not affect its rescuing efficacy. These findings suggest that there is something making H3.3 intrinsically different from canonical H3.2. Indeed, the authors indicate that the presence of a serine at position 31 in H3.3, a residue that is phosphorylated during mitosis is critical in regulating its developmental role. Mutating S31 to aspartate, a phosphomimetic residue is sufficient to rescue the phenotypes caused by H3.3 loss. This suggests that dynamics of phosphorylation of this residue might not be essential, whilst what it is relevant is its charge and/or proteins reading it.

Overall, this is a sound and important study with potential implications for our understanding of this chromatin remodelling mechanism in physiology and disease. A number of studies have proposed a model whereby dynamics of H3.3 loading are the critical factor underlying its role in chromatin regulation. However, it was unclear that this was the whole story. Indeed, the present manuscript suggests that H3.3 intrinsic features play an important role, specifically the presence of S31. I am convinced that this work is worth reporting and of appeal to a varied audience. However, I have a number of comments that need to be addressed in order to strengthen its conclusions and impact.

- The authors suggest several mechanisms that could underlie the importance of S31. However, I am convinced that the authors should attempt to address at least a couple of these hypotheses. In this respect, the idea that S31 can create a docking site for readers, such as BS69 is interesting. The authors should attempt identifying interacting proteins binding H3.3 in a S31 dependent manner using the mutants reported here (using affinity purification/mass spec). Alternative approaches can be proposed, but I strongly believe more insights into the potential role of S31 as docking site for chromatin remodellers should be produced.
- Affinity purification experiments could also provide insights into what kinases interact with h3.3 in the experimental model used in this manuscript.
- Another possibility is that S31 could influence neighbouring histone marks. Have the authors studied whether N-tail modifications are affected at global levels depending on the type of S31 mutations (to A or D)? Mass spec for histone marks could be attempted.
- The authors propose that the incorporation pathway of H3.3 does not really matter in regulating its function in development. This is really intriguing, but I believe discussion on this should be expanded, as authors fail to speculate on reasons underlying these findings, especially since the literature suggests that dynamics of loading is a critical factor. In this respect, one could speculate that interaction of H3.3 with a critical factor/reader prior to chromatin incorporation and irrespective of the type of chaperone is what counts.

We thank all three reviewers for their interest in on our manuscript and all their constructive feedback. This was a real incentive to deepen our study in order to consider their major point, which is to be able to provide further mechanistic insights into the role and regulation of H3.3S31 phosphorylation. To achieve this, we carried out a number of new key experiments, and as you will see, this has significantly improved the paper and strengthened the overall work. This not only documented the observation of a need for a specific residue in H3.3 at a given time but also how it is needed at this time and also how it can be regulated.

To illustrate this, I would like to stress three sets of new data enabling us to bring strong arguments to address mechanistically the role and regulation of H3.3S31 phosphorylation:

- 1) Concerning regulation of H3.3S31ph: we assessed H3.3S31ph dynamics exploiting the *in vitro* system in *Xenopus* egg extracts and tested a series of candidate kinases and phosphatases potentially associated with H3.3S31ph dynamics. We found that H3.3S31ph occurs later than some other known phosphorylation on histone H3 and that the classical mitotic network involving both Aurora B and Chk1 kinases impacted the mitotic phosphorylation H3.3S31ph. However, its dephosphorylation was not dependent on the PP1/PP2, which removed all other tested phosphorylation. These data are presented in Fig. 4 panel C and panel D.
- 2) Concerning mechanistic role part 1: We identified by mass spectrometry key factors either attracted or repulsed by H3.3S31ph using peptides corresponding to H3 N-terminal tails carrying mutations for H3.3S31 after incubation in *Xenopus* egg extracts. A complete list is provided. Using the top scores from gene ontology, we could highlight that H3.3S31D (negative charge) could attract in interphase proteins involved in regulating transcription (β -Catenin pathway) and in mitosis repulsed factors involved in chromosome condensation and transcription splicing. These important findings are reported in a new Fig. 6 (along with a supplementary for a comprehensive report). This can explain how the change in H3.3 can contribute in engaging/ sustaining developmental transition that are critical after the MBT and can impact cell cycle.
- 3) Concerning mechanistic role part 2: We provide key information concerning how H3.3S31ph crosstalk with neighboring residues and their modifications. For this, we generated new stable cell lines expressing H3.3S31 mutants and could examine changes associated to endogenous H3 and to the tagged-H3 mutants. We then confirmed in *Xenopus* embryos our finding, examining the status of H3 in the embryo on both endogenous and exogenous tagged versions. A common salient feature that came out was a crosstalk *in cis* with H3.3K27 with H3.3S31D highlighting the capacity to promote a chromatin opening/ permissive state. These data are presented in a new Fig. 7 along with its corresponding supplementary for details.

In addition, we completed our experiments and deepened all the analyses as requested by introducing all controls requested. Furthermore, we carefully edited our text to explain along the new findings.

We believe that the revised manuscript is much improved and hope that it can now be accepted for publication.

List of changes to figures:

Figure 1: Former panel B has been moved to Supplemental figure 1 A

Figure 2: We fused Figure 2 with Figure 3 to show side-by-side all mutants and appropriated controls

Figure 3: New Panel A provides results for new immunoprecipitation with H4, DAXX and p150

Figure 4:

- We moved former panel C to Supplemental figure 4 C
- New panel C to show the phosphorylation dynamics of H3 (T3, S10, S28) and H3.3S31 in *Xenopus* egg extracts, and in a new panel D with various kinase and phosphatase inhibitors

Figure 5: New panel B provides new immunoprecipitation with H4, DAXX and p150

Figure 6 is new

- New panel A shows a scheme for the strategy using peptides, harboring various substitutions at S31, incubated in egg extracts to pull down partners identified by mass spectrometry analysis
- New panel B shows Venn diagrams of proteins associated to each condition.
- New panel C lists the factors (associated or repulsed) using gene ontologies for H3.3S31D

Figure 7 is new

- New panel A shows a scheme depicting H3.3S31 and key neighboring PTMs
- New panel B identifies how H3.3S31 negative charge crosstalked with neighboring residues in stable human cells expressing transgenic H3.3 with various substitutions at S31.
- New panel C shows the same type of analysis for H3.3S31 crosstalk in developing embryos.

Figure 8 is former Figure 7

Supplemental figure 1

- Panel A corresponds to former Figure 1 B
- Former panel B has been updated for the morpholino design and moved to material and methods

Supplemental figure 2 and 3 are fused to show side-by-side all mutants and appropriated controls

- New panel B shows now new fractionations including α -tubulin and H4 as controls for soluble and chromatin fractions, respectively.

Supplemental figure 3: A new panel B corresponds to a new set of immunoprecipitation *in vivo* corresponding to H3.3 dedicated histone chaperones

Supplemental figure 4

- Panel C corresponds to former Figure 5 C
- New panel D shows now a new set of experiment using stable human cells expressing transgenic H3.3 with various substitutions at S31 treated with various kinase inhibitors

Supplemental figure 5: A new panel B shows now a new set of fractionations including α -tubulin and H4 as controls for soluble and chromatin fractions, respectively

Supplemental figure 6 is new

- New panel A shows the phosphorylation state of each peptide in interphase and mitotic extracts
- New panel B lists gene ontology associated to each peptide in interphase and mitotic extracts
- New panel C corresponds to gene ontologies for proteins binding to both H3.3S31 and H3.3 S31A peptides

Supplemental figure 7 is new

- New panel A corresponds to the generation of stable human cells expressing transgenic H3.3 with various substitution at S31
- New panel B corresponds to cell cycle analysis of the new stable cell lines
- New panel C shows the characterization of related proteins after expression of H3 constructs
- New panel D displays a schematic model of the crosstalk

Supplemental figure 8 is new to show H3 PTM antibody crosstalk with unmodified H3 peptides

Note: In each figure, molecular weights and quantifications are indicated as appropriate.

Reviewers' comments:

- Reviewer #1 (Remarks to the Author):

The manuscript "Regardless of the deposition pathway, amino acid 31 in histone variant H3 is essential at gastrulation in *Xenopus*" by Sitbon et al. is an insightful study investigating functional consequences in *Xenopus* development by mutating H3.3 specific residues. The main and surprising discovery is the observation that amino acids required for H3.3-specific chaperone binding and subsequent variant chromatin deposition are not required for embryonic frog development. Yet, the unique serine 31 that has been found to be phosphorylated during mitosis is essential, most likely in a phosphorylated, negatively charged manner. These findings are important and novel, unfortunately, experiments investigating the mechanism by which amino acid 31 contributes to proper embryonic development are missing. It is very nice descriptive study, but at least one experimental hint towards the mechanism should be provided. My major and minor concerns are listed in detail below:

We appreciate that the reviewer acknowledges that our findings are important and novel. In our revised manuscript, we have now addressed his/her request for at least one experimental hint towards a mechanism/regulation along with her/his specific comments. We now document the involvement of mitotic kinases, including Chk1 and Aurora B kinases, for H3.3S31ph. Furthermore, we show that the presence/absence of H3.3S31 negative charge specifically alters the capacity to bind/repulse particular factors and impact in cis neighboring PTM, modifications known for their importance in transcription regulation. The loss of H3.3S31ph by inducing local changes in the chromatin landscape (binders and PTMs) does now provide a molecular explanation for the developmental failure observed after depletion of H3.3. We thus hope that this significantly improved version of the manuscript with the important set of new experimental data will satisfy the reviewer.

Major concerns:

1. The authors state that an H3F3B morpholino was used. Did the authors also test an H3F3A morpholino and do they know anything about the expression of both H3.3 genes during *Xenopus* development? Did the author test and verify H3.3 knockdown efficiency with immunoblots?

*Indeed, the expression of H3.3 genes during *Xenopus* early development has been previously documented in our laboratory¹ (See annex, Figure 1). This is consistent with the database of mRNA and protein expression for vertebrate embryogenesis from Mark Kirschner lab² (available at: http://kirschner.med.harvard.edu/mz_site/blast/index4.html). We now provide this information in the method section to explain the choice of the morpholino that targets the gene b, because it shows a major peak of expression at gastrulation compared to all other H3.3 genes. We also refer more clearly in the text to our previous publication³, in which we had already demonstrated the specificity of the morpholino against H3.3 (See Annex, Figure 2). Moreover, we have also discussed the design and specificity of morpholino in the main text and the material and methods.*

2. Is the same phenotype observed in other *Xenopus* species?

While this is an interesting point, we would be opened to explore this in the future in collaboration with other laboratories. Indeed, access to other species for experimentation is not trivial both for regulatory rules in animal experimentation and for the hosting, as well as their suitability to the experiments at stake.

3. Do the authors find more apoptotic cells in their H3.3 depleted embryos? The movies somehow suggest this.

This is indeed the case, as previously observed by TUNEL assay in H3.3 depleted embryos³ (See Annex, Figure 3). We now mention this in the text.

4. Figure 2B: The quantification of the rescue with WT H3.3 is not shown. As the rescue seems to reach around 70% (the scaling is not detailed enough), it would be good to compare this number to the WT control. The rescue seems high, but it is not clear whether the single chaperone binding site mutants are as good as WT H3.3 in rescuing the phenotype in this particular experiment. Additionally, have the authors looked at later stages? Is the development to a tadpole normal or are there problems?

We have now combined Figure 2 and 3 and show the controls not only the rescue with WT H3.3 but also for the failure with WT H3.2. We have also improved the presentation of the quantification in order to facilitate the comparison. Concerning the follow up at later stages, we should stress that the concentrations of morpholino and mRNA that we injected have been optimized for gastrulation rescue, however the efficiency of the rescues drops at later stage due to the transient nature of the expression system. This is not surprising given the lower stability of mRNAs compared to morpholinos. Thus, we would not be confident to draw conclusions concerning the phenotype observed for later stages with these settings. We have added this explanation in the material and methods as well as in the discussion.

5. Figure 4A: The authors did not test whether eH3.3 SVM is still able to bind DAXX/ATRX. Most likely not, but it should be tested, as it might explain the rescue ability of this mutant.

The reviewer is correct and this missing piece of information is indeed relevant. We have now carried out a new set of immunoprecipitation experiments in Xenopus egg extracts, the same type of extracts used to assess the mode of incorporation. Consistent with our hypothesis, we do find that DAXX binding is altered when the H3.3 AIG motif is swapped with the one of H3.2. Furthermore, in line with comments from reviewer 2, we have also added additional controls including p150, another subunit of CAF-1, the dedicated histone chaperone for the replicative variant H3.2 in Xenopus. We now show these data in new panels in Figure 3 (Figure 3 A) and Figure 5 (Figure 5 B). In addition, we have performed new immunoprecipitation in developing embryos to confirm in vivo those findings (Supplemental figure 3 B). We adjusted the text accordingly.

6. Figure 4: It would be important to perform CHIP-seq (or CHIP-qPCR in case specific sites are already known) with the different mutants to show that they are deposited into different regions than WT H3.3 (including heterochromatin regions) and are still able to rescue the morpholino phenotype. Just the demonstration of generalized chromatin deposition is not sufficient.

Since histone distribution has been mostly studied in cell lines and the situation during early development is not documented, this is indeed an interesting aspect opening a whole range of experiments and new findings. This is not simply a control for our experiments but given the scope of the work, it represents a project on its own that would per se deserve a complete comprehensive study. In the present study, we wished first to show that swapping histone chaperone recognition motif in histone variants was sufficient to change histone chaperone binding and interactions.

This in turn allows modifying the mode of incorporation into chromatin of these histone mutants. Whether a change in histone location ensues or not is certainly an interesting matter, and we now discuss this aspect in the discussion to open future avenues.

7. If technically possible, the authors should provide anti-H3.3S31ph staining of different embryonic developmental stages to determine the exact appearance and tissue localization of this modification. Is H3.3 S31 phosphorylated around blastopore stage, explaining the observed lethality of S31A mutants at this particular time during development?

In order to explore H3.3S31ph occurrence throughout development in mitosis, we collected Xenopus embryos at different stages of early development and prepared corresponding whole embryo extracts (See annex, Figure 4). The low numbers of nuclei in embryos at early developmental stages did not allow detecting easily H3.3S31ph for pre-MBT by Western blot. We thus used immunohistochemistry after nocodazole treatment (to block cells in metaphase) and revealed H3.3S31ph at all cell stages that we analyzed without any particular specific tissue distribution or development stages. At the moment, the data are provided for reviewers only and we could add them if considered useful.

8. The main and most important question is about the function of H3.3S31 (and its phosphorylation) during development. Why is it so important regardless of its site of deposition? I am missing experiments trying to decipher the mechanism behind H3.3S31ph function in development. The authors raise several interesting ideas in the discussion section (role of H3.3S31ph during cell cycle, in genome stability, for CENP-A incorporation, for transcription regulation...), but do not experimentally address them. It should be possible to look at cell cycle distributions of mutant embryos, the ploidy and whether CENP-A is incorporated at centromeres. Importantly, the authors should look into transcriptome changes and whether those might provide a clue into the process that went wrong in S31A mutants.

Our initial objective was to focus on showing that at this time of development, H3.3 functions do not rely on the mode of deposition but on the nature of the variant itself. Thus, while we have decided to remain focused on this particular aspect, we also wished to understand from where the necessity to have H3.3S31 arises. We have thus carried out two important sets of additional studies. First, we used peptides corresponding to the N-terminal tail of H3 carrying different mutations for the residue at position 31 and incubated them with mitotic or interphase extracts from Xenopus eggs. This allowed us to retrieve specific binding partners that we could then identify by mass spectrometry analysis (Figure 6). The data obtained provide an important source of information that we now share. We also stress some key aspects related to attraction/repulsion for the H3.3S31 negative charge. The factors identified allow us to make a link with transcription, chromosome compaction and mRNA splicing. Second, we explored further the PTM landscape influenced by the presence/absence of the charge/modification at position 31 (Figure 7). Our data enable us to show a specific crosstalk between the H3.3S31 negative charge and neighboring PTMs, using both new stable cell lines and developing embryos. Most remarkably, we find that H3.3S31D (constitutive charge) shows an increase in H3.3K27ac. This is in agreement with a study published recently showing that H3.3S31ph could stimulate H3.3K27ac in mESC⁴. Notably, our data further enabled to have a resolution in cis, and we could show this in vivo using developing embryos at gastrulation. Interestingly, we could not observe the crosstalk with H3.3K36me3 found in activated macrophages, as reported in a prepublication⁵. These new pieces of information and recent publications help us to put forward the importance of the crosstalk between H3.3S31 and H3.3K27 PTMs.

We now discuss these data and their relevance in the context of gastrulation and the major transcriptional program changes at this stage of development.

Minor:

1. The authors talk about "DNA synthesis coupled (DSC)" or "DNA synthesis independent (DSI)" mechanisms. In the large majority of papers on histone variants the phrases "replication dependent (RD)" or "replication independent (RI)" have been established. It would be preferable to switch back to RD or RI throughout the text.

While we do appreciate that the literature does often use "replication dependent (RD)" or "replication independent (RI)", this is in fact not formally correct. Indeed, we do know that incorporation due to DNA synthesis in the context of DNA repair (independently of replication) also leads to incorporation of replicative variants^{6,7}. Thus, we favor to refer to "DNA synthesis coupled (DSC)" and "DNA synthesis independent (DSI)" pathways to reflect these aspects, as previously explained here⁸. We have now stated more clearly the rationale in the text, since this is largely overlooked in the chromatin community, and this may help for the future.

2. Figure 1A and B are largely redundant and have been depicted in many other publications. I suggest moving one of these Figures into the Supplement.

Although these aspects have been already presented in other publications separately, given their importance for the understanding of the paper by a general audience, they remain critical. Thus, we have now moved Figure 1 B in the Supplemental figure 1 (Supplemental Figure 1 A), as suggested by the reviewer.

3. I am missing molecular weight indications next to immunoblots.

We have now added to all western blots the corresponding molecular weight.

4. Legends for Figures 2B, 3, 5A, and 6A: Indicate numbers of embryos analyzed and whether error bars are calculated as SD or SEM.

We have now added in the corresponding figure legends the number of embryos analyzed and that we measured standard deviation.

5. Supplemental Figures 2B and 3B: Control antibodies to show efficient and specific fractionation are missing. (Remove one "n" from "fractionation" in the figure).

To provide more controls to better show efficiency and specificity of the fractionation, we have performed another set of fractionations. We have now included in Supplemental figure 2 (Supplemental figure 2 B) and Supplemental figure 5 (Supplemental figure 5 B) new H4 and α -tubulin controls for soluble and chromatin fractions, respectively. We also removed the extra "n" from fractionation in the corresponding figures.

6. Nomenclature: please use the term "H3.3S31ph" instead of "H3.3S31p". It looks too similar to a mutation to a proline.

We have replaced all "H3.3S31p" by "H3.3S31ph" in all texts and figures.

- Reviewer #2 (Remarks to the Author):

In this manuscript, Sitbon et al. test the *Xenopus laevis* developmental function of histone H3.3 variant amino acids. Other than altered chaperone binding and deposition timing, the functional significance of the H3.3 amino acid differences from deposition-dependent H3.2 has been enigmatic in all model systems probed. Therefore, this study, employing *Xenopus* embryos, cell-free extracts, and cultured cells, is informative. Overall, this is a well-executed and concise study, generally supporting the conclusion that S31 phosphorylation is essential for *Xenopus laevis* gastrulation. Additionally, the authors show clear evidence that altering the chaperone-mediated deposition pathway does not influence viability (which is rather remarkable, and contrary to much of the collected wisdom of the last 10 years or so on these variants). The authors identify a striking role for acidic character at residue H3.3S31 that does not need to be dephosphorylated for the embryo to proceed through gastrulation. However, the authors do not test what the role of this phosphorylation may be, nor do they propose any experiments to decipher this question. While this work should be published, I strongly recommend that the authors consider additional experiments to answer what biological or biochemical function H3.3S31ph is performing. I have a few general and specific comments.

We thank the reviewer for acknowledging that our work should be published. He/she asks for more biological or biochemical experiments to reveal the functions of H3.3S31ph. We have improved the revised manuscript accordingly. We now document the involvement of mitotic kinases, including Chk1 and Aurora B kinases, for H3.3S31ph. Furthermore, we show that the presence/absence of H3.3S31 negative charge specifically alters the capacity to bind/repulse particular factors and impact in cis neighboring PTM, modifications known for their importance in transcription regulation. The loss of H3.3S31ph by inducing local changes in the chromatin landscape (binders and PTMs) does now provide a molecular explanation for the developmental failure observed after depletion of H3.3. We hope that this revision including new experimental works addressing the mechanism will satisfy this reviewer.

1. No specific evidence is shown of H3.3 morpholino knockdown, only the rescues are shown. In particular, the morpholino titration (Fig S1C) would be more compelling—and rule out off target effects—with direct evidence of corresponding loss of H3.3 protein. As the authors have a general H3 and an H3.3 specific antibody, this should be straightforward (as done either by TAU or western blot in the prior Almouzni lab paper Szenker et al 2012).

This is in line with comments from reviewer 1. As previously explained, we had already extensively characterized the use of this morpholino, especially regarding its specificity³ (See Annex, Figure 2). We have now explained those results in the current version of the manuscript. In addition, we have also updated the design and specificity of morpholino panel and moved it in the material and methods.

2. In Supplemental Fig 2, it appears that there may be substantial soluble (nuclear or cytoplasmic?) H3, but not soluble H4. This is curious, especially at Stage 12 embryos. Are these cytoplasmic histones bound to chaperones? As both the endogenous (Control) and rescued embryos contain soluble histone (and HA tagged histone), this suggests a bona fide biological role. Titrating the dose of rescue mRNA could address this question.

We repeated these experiments and blot H4 with another H4 antibody that gives a higher signal and could thus readily detect H4 in the soluble pool using this new antibody. The previous observations are likely due to the low antibody sensitivity (notorious for H4).

These sets of experiments have now been updated in Supplemental figure 2 (Supplemental figure 2 B) and Supplemental figure 5 (Supplemental figure 5 B).

3. In the cell-free extract experiments, the authors should also blot for H4 (and either H2A or H2B as a negative control). H4 enrichment in the tagged H3.2 and H3.3 will ensure that it is truly binding to chaperones as a native dimer or tetramer complex. In Fig 4B, the aphidicolin experiment is cleverly designed to demonstrate replication-independent deposition.

In line with reviewer 1, we agree with the reviewer and added these extra controls to strengthen our conclusion. Therefore, we have carried out a new set of immunoprecipitation experiments in Xenopus egg extracts, including p150 and DAXX histone chaperones. We have also blotted the new immunoprecipitation experiments with H4 and found it specifically associated to the exogenous histones. We now show these new panels in Figure 3 (Figure 3 A) and Figure 5 (Figure 5 B). The text has been adjusted accordingly. Finally, it was satisfying to note that the reviewer found the experiment cleverly designed.

4. Since the modest prior literature on H3.3S31ph and Fig 5 shows that it has different kinetics than H3S10ph (e.g. it appears later in metaphase), I recommend using cycling egg extract for a fine-grained comparison of the two marks. In particular, the extract can be supplemented with a high concentration of purified recombinant H3.3S31A and H3.3S31D (in complex with H4) to ask where in mitosis the mutation elicits a defect. This may permit insight into the specific mitotic timing of the gastrulation defect (better than the correlation with the analysis of A6 cells).

We appreciate the suggestion of the reviewer to better define the dynamics of H3.3S31ph compared to H3S10ph. We have extended our analysis regarding the dynamics of the marks. Regarding solely the kinetics, we analyzed the main phosphorylation of H3, which are H3T3ph, H3S10ph, H3S28ph and H3.3S31ph. Furthermore, and consistent with our data obtained by immunofluorescence on cell lines, we could detect a peak for H3S10ph earlier than for H3.3S31ph (Figure 4 C). We believe that this provides a strong argument for specificity concerning H3.3S31ph compared to H3S10ph. Moreover, the use of the extract could be insightful both to further explore dedicated kinases and assess potential cell cycle defects on exogenous nuclei. We indeed used it to explore the kinase inhibitors (next point). However, to examine chromosomal defects, one should note that Xenopus sperm chromatin retains H3-H4 nucleosomes⁹ (See Annex, Figure 5). In addition, levels of recombinant H3.3 to incorporate into chromatin will be very difficult to attain with the limited number of divisions that can be achieved in vitro with these extracts. We could thus envisage to possibly explore these aspects in the future using mouse sperm nuclei that replace H3 by protamines, but this adds another layer of complexity that can obscure the interpretation. Of note, we also provide an extensive characterization of the H3 PTM antibodies used in the present studies.

5. Similarly, identification of the kinase responsible for the phosphorylation in Xenopus would help understand the function. Using either a Chk1 or Aurora B inhibitor in the extract experiments would help answer this question, and permit more functional understanding.

We thank the reviewer for this very useful suggestion that prompted us to get into more functional aspects. We have tested Chk1 and Aurora B inhibitors in order to identify the kinases (Figure 4 D). We used Xenopus egg extracts treated with these different inhibitors at different concentrations and revealed in all cases a decrease for H3 phosphorylation.

This suggests that these kinases, involved in the same pathways^{10,11}, are both involved directly or indirectly in these phosphorylation. This may in fact explain the current discrepancy in the literature. Moreover, we also confirmed these results using human cells by immunofluorescence. We felt that it was important to examine most of H3 phosphorylation in parallel to actually delineate some level of specificity for each individual residue. Concerning the phosphatases, we inhibited the PP1/PP2A, which affected all H3 phosphorylation analyzed as expected, but surprisingly not H3.3S31ph. This suggests either another phosphatase or an active eviction of H3.3S31ph as a mechanism to remove this modification. We have now assembled the data in Figure 4 and Supplemental figure 4 and expanded the corresponding part of the text and discussion.

- Reviewer #3 (Remarks to the Author):

The work by Sitbon et al. investigates the role of the histone H3.3 variant in regulating developmental mechanisms in *Xenopus*. Incorporation of H3.3 is emerging as fundamental mechanism in chromatin remodeling, and genetic alterations affecting H3.3 and/or its deposition machinery have been reported in multiple human cancers. It is therefore important to dissect both function and regulation of this variant as this could provide key insights into pathogenesis of several neoplasms. In this manuscript, the authors report that the ability of H3.3 to regulate development is independent from its loading mechanisms, as mutating the chaperone-interacting motif of H3.3 to the one carried by the S-phase restricted H3.2 variant does not affect its rescuing efficacy. These findings suggest that there is something making H3.3 intrinsically different from canonical H3.2. Indeed, the authors indicate that the presence of a serine at position 31 in H3.3, a residue that is phosphorylated during mitosis is critical in regulating its developmental role. Mutating S31 to aspartate, a phosphomimetic residue is sufficient to rescue the phenotypes caused by H3.3 loss. This suggests that dynamics of phosphorylation of this residue might not be essential, whilst what it is relevant is its charge and/or proteins reading it. Overall, this is a sound and important study with potential implications for our understanding of this chromatin remodeling mechanism in physiology and disease. A number of studies have proposed a model whereby dynamics of H3.3 loading are the critical factor underlying its role in chromatin regulation. However, it was unclear that this was the whole story. Indeed, the present manuscript suggests that H3.3 intrinsic features play an important role, specifically the presence of S31. I am convinced that this work is worth reporting and of appeal to a varied audience. However, I have a number of comments that need to be addressed in order to strengthen its conclusions and impact.

We are pleased to see that the reviewer states that our study is important and have potential implications for physiological and pathological contexts. We are confident that addressing his/her comments has indeed improved substantially our work. We hope that this revised manuscript will now satisfy this reviewer.

1. The authors suggest several mechanisms that could underlie the importance of S31. However, I am convinced that the authors should attempt to address at least a couple of these hypotheses. In this respect, the idea that S31 can create a docking site for readers, such as BS69 is interesting. The authors should attempt identifying interacting proteins binding H3.3 in a S31 dependent manner using the mutants reported here (using affinity purification/mass spec). Alternative approaches can be proposed, but I strongly believe more insights into the potential role of S31 as docking site for chromatin remodelers should be produced.

We agree with the reviewer that performing mass spectrometry could give us valuable insights into the functions of the specific H3.3S31ph mark. We used the egg extract system, both in interphase and in metaphase, combined with biotinylated peptides corresponding to the different mutant H3 N-terminal tails (Figure 6). The data obtained provide an important source of information that we now share. We also stress some key aspects related to attraction/repulsion for the H3.3S31 negative charge. The factors identified allow us to make a link with transcription, chromosome compaction and mRNA splicing. Interestingly, the splicing repulsion by H3.3S31D is in agreement with previous studies that have shown that H3.3S31ph would prevent the binding to H3.3K36me3 of ZMYND11, involved in intron retention^{12,13}. Since this negative charge affected mechanisms associated to specific PTMs (such as splicing and H3K36me3), we analyzed its impact on the PTM landscape and found that it would increase H3.3K27ac in cis (Figure 7), which we discuss below in point 3.

2. Affinity purification experiments could also provide insights into what kinases interact with h3.3 in the experimental model used in this manuscript.

Since the dynamics of kinases and phosphatases is difficult to detect by mass spectrometry, as it is highly transient per se, we instead used an approach combining egg extracts and inhibitors in order to examine the main enzymes. This is also in lines with experiments proposed by the other reviewers. We have used Chk1 and Aurora B inhibitors in different contexts in order to identify the kinases. We used Xenopus egg extracts treated with the different inhibitors as well as with different concentrations and revealed in all cases a decrease for both H3S10ph and H3.3S31ph. This suggests that these kinases, involved in the same pathways^{10,11}, are both responsible for these phosphorylation, allowing us to resolve the current discrepancy in the literature. We have added this data in a new panel in Figure 4 and Supplemental figure 4 and commented the results in the revised version of this manuscript.

3. Another possibility is that S31 could influence neighboring histone marks. Have the authors studied whether N-tail modifications are affected at global levels depending on the type of S31 mutations (to A or D)? Mass spec for histone marks could be attempted.

Indeed, the fact that H3.3S31ph could crosstalk with other histone marks is an appealing possibility. We have thus investigated potential crosstalk by setting up a new stable cell lines expressing the different H3.3S31 mutants (Figure 7). We could then immunoprecipitate the different exogenous forms and analyzed their associated PTMs. We could not detect changes for H3.3K36me3 as observed for activated macrophages in another recent prepublication⁵. However, and most remarkably, we find that H3.3S31D (constitutive charge) shows an increase in H3.3K27ac in the new cell lines expressing H3 constructs. We also found this enrichment of H3.3K27ac in vivo with the developing embryo at gastrulation, in line with recent data obtained in mESC⁴. Notably, the design of our experiment allows to obtain a resolution in cis. This enables us to discuss in the text how H3.3S31ph can alter neighboring residue and their post-translational marks to impact genome function. This is critical to consider, regarding the importance of inducing and maintaining new transcriptional programs at this time of development.

4. The authors propose that the incorporation pathway of H3.3 does not really matter in regulating its function in development. This is really intriguing, but I believe discussion on this should be expanded, as authors fail to speculate on reasons underlying these findings, especially since the literature suggests that dynamics of loading is a critical factor.

In this respect, one could speculate that interaction of H3.3 with a critical factor/reader prior to chromatin incorporation and irrespective of the type of chaperone is what counts.

We have extended the discussion considering the new experiments and also speculated further in order to stimulate the scientific community that should be able to build on our findings and their implications.

Annex

Szenker, Thesis manuscript, 2012

Figure 1: H3.2 and H3.3 mRNA expression levels through development. Analysis of the transcripts encoding H3.2 (a and b genes) and H3.3 (a, b, c et d genes) by RT-qPCR from total mRNA extracted from embryos at different developmental stages. Each error bar represents expression for each transcript, normalized with the ODC housekeeping gene in arbitrary unit (a.u.). For a better comparison, values from the blastula stage have been fixed to 1. Adapted from Figure 48 A from E. Szenker PhD manuscript (available here: <https://tel.archives-ouvertes.fr/tel-00836233>)

Szenker et al., 2012

Figure 2: Specificity of H3.3 and H3 morpholinos. Soluble proteins extracted from embryos injected with different morpholino. Results used a 2-fold dilution series (gradient bar) and detection with indicated antibodies. Anti-β actin and memcode staining served as loading controls. Adapted from Figure 1 A from Szenker et al., 2012.

Szenker et al., 2012

Figure 3: TUNEL assay of morpholino treated embryos. We injected the indicated MO (4.6 ng) in one cell of 2-cell stage embryos and performed a TUNEL assay when controls embryos reached the neurula stage. White arrows indicate the TUNEL positive cells in the injected side of H3.3 morphants. The majority of white apoptotic cells comes off the embryos at the beginning of the experimental procedure that involves dechorionization. Adapted from Figure 4 C from Szenker et al., 2012.

H3.3S31ph staining

Figure 4: H3.3S31ph levels through *Xenopus* early development. *A)* Whole embryo extracts from different developmental stages for H3.3S31ph, as well as other histones. Each sample corresponds to a loading of 48ng and 72ng of proteins from extracts. *B)* Immunohistochemistry for H3.3S31ph. Embedded embryos treated with nocodazole are sliced and used to observe in vivo H3.3S31ph. Scale bar corresponds to 200 μ m.

H3.3 retention

Figure 5: H3.3 retention in *Xenopus* sperm chromatin. Purified chromatin from sperm nuclei or/and remodeled into the mitotic extract.

References

- 1 Szenker, E. PhD Manuscript. (2012).
- 2 Peshkin, L. *et al.* On the Relationship of Protein and mRNA Dynamics in Vertebrate Embryonic Development. *Dev Cell* **35**, 383-394, doi:10.1016/j.devcel.2015.10.010 (2015).
- 3 Szenker, E., Lacoste, N. & Almouzni, G. A developmental requirement for HIRA-dependent H3.3 deposition revealed at gastrulation in *Xenopus*. *Cell Rep* **1**, 730-740, doi:10.1016/j.celrep.2012.05.006 (2012).
- 4 Martire, S. *et al.* Phosphorylation of histone H3.3 at serine 31 promotes p300 activity and enhancer acetylation. *Nat Genet* **51**, 941-946, doi:10.1038/s41588-019-0428-5 (2019).
- 5 Armache, A. *et al.* Phosphorylation of the ancestral histone variant H3.3 amplifies stimulation-induced transcription. *bioRxiv*, doi:doi.org/10.1101/808048 (2019).
- 6 Polo, S. E., Roche, D. & Almouzni, G. New histone incorporation marks sites of UV repair in human cells. *Cell* **127**, 481-493, doi:10.1016/j.cell.2006.08.049 (2006).
- 7 Gaillard, P. H. *et al.* Chromatin assembly coupled to DNA repair: a new role for chromatin assembly factor I. *Cell* **86**, 887-896 (1996).
- 8 Ray-Gallet, D. & Almouzni, G. DNA synthesis-dependent and -independent chromatin assembly pathways in *Xenopus* egg extracts. *Methods Enzymol* **375**, 117-131, doi:10.1016/s0076-6879(03)75008-3 (2004).
- 9 Katagiri, C. & Ohsumi, K. Remodeling of sperm chromatin induced in egg extracts of amphibians. *Int J Dev Biol* **38**, 209-216 (1994).
- 10 Mackay, D. R. & Ullman, K. S. ATR and a Chk1-Aurora B pathway coordinate postmitotic genome surveillance with cytokinetic abscission. *Mol Biol Cell* **26**, 2217-2226, doi:10.1091/mbc.E14-11-1563 (2015).
- 11 Kabeche, L., Nguyen, H. D., Buisson, R. & Zou, L. A mitosis-specific and R loop-driven ATR pathway promotes faithful chromosome segregation. *Science* **359**, 108-114, doi:10.1126/science.aan6490 (2018).
- 12 Guo, R. *et al.* BS69/ZMYND11 Reads and Connects Histone H3.3 Lysine 36 Trimethylation-Decorated Chromatin to Regulated Pre-mRNA Processing. *Mol Cell* **56** (2014).
- 13 Wen, H. *et al.* ZMYND11 links histone H3.3K36me3 to transcription elongation and tumour suppression. *Nature* **508**, 263-268, doi:10.1038/nature13045 (2014).

REVIEWERS' COMMENTS:

Reviewer #2 (Remarks to the Author):

The authors have satisfactorily addressed my critique.

Reviewer #3 (Remarks to the Author):

The Authors have addressed most of my points. I am convinced the manuscript has been further improved. Importantly, the new data have added key mechanistic insights into the regulation and role of the specific histone residue.